# Metalearning to Continually Learn In Context

## Abstract

General-purpose learning systems should improve themselves in open-ended fash-
ion in ever-changing environments. Conventional learning algorithms for neural
networks, however, suffer from catastrophic forgetting (CF)—previously acquired
skills are forgotten when a new task is learned. Instead of hand-crafting new
algorithms for avoiding CF, we propose Automated Continual Learning (ACL) to
train self-referential neural networks to meta-learn their own *in-context* continual
(meta-)learning algorithms. ACL encodes continual learning desiderata—good
performance on both old and new tasks—into its meta-learning objectives. Our
experiments demonstrate that, in general, in-context learning algorithms also suffer
from CF but ACL effectively solves such "in-context catastrophic forgetting". Our
ACL-learned algorithms outperform hand-crafted ones and popular meta-continual
learning methods on the Split-MNIST benchmark in the replay-free setting, and
enables continual learning of diverse tasks consisting of multiple few-shot and stan-
dard image classification datasets. Going beyond, we also highlight the limitations
of in-context continual learning, by investigating the possibilities to extend ACL to
the realm of state-of-the-art CL methods which leverage pre-trained models.[1]

## 1 Introduction

Enemies of memories are other memories [1]. Continually-learning artificial neural networks (NNs)
are memory systems in which their *weights* store memories of task-solving skills or programs, and
their *learning algorithm* is responsible for memory read/write operations. Conventional learning
algorithms—used to train NNs in the standard scenarios where all training data is available *at once*—
are known to be inadequate for continual learning (CL) of multiple tasks where data for each task
is available *sequentially and exclusively*, one at a time. They suffer from "catastrophic forgetting"
(CF; [2–5]); the NNs forget, or rather, the learning algorithm erases, previously acquired skills, in
exchange of learning to solve a new task. Naturally, a certain degree of forgetting is unavoidable
when the memory capacity is limited, and the amount of things to remember exceeds such an upper
bound. In general, however, capacity is not the fundamental cause of CF; typically, the same NNs,
suffering from CF when trained on two tasks sequentially, can perform well on both tasks when they
are jointly trained on the two tasks at once instead (see, e.g., [6]).

The real root of CF lies in the learning algorithm as a memory mechanism. A "good" CL algorithm
should preserve previously acquired knowledge while also leveraging previous learning experiences
to improve future learning, by maximally exploiting the limited memory space of model parameters.
All of this is the *decision-making problem of learning algorithms*. In fact, we can not blame the
conventional learning algorithms for causing CF, since they are not aware of such a problem. They
are designed to train NNs for a given task at hand; they treat each learning experience independently
(they are stationary up to certain momentum parameters in certain optimizers), and ignore any

---

[1]Here we'll add a link to our public GitHub code repository.

potential influence of current learning on past or future learning experiences. Effectively, more sophisticated algorithms previously proposed against CF [7, 8], such as elastic weight consolidation [9, 10] or synaptic intelligence [11], often introduce manually-designed constraints as regularization terms to explicitly penalize current learning for deteriorating knowledge acquired in past learning.

Here, instead of hand-crafting learning algorithms for continual learning, we train self-referential neural networks [12, 13] to meta-learn their own "in-context" continual learning algorithms. We train them through gradient descent on learning objectives that reflect desiderata for continual learning algorithms—good performance on both old and new tasks, including forward and backward transfer. In fact, by extending the standard settings of few-shot or meta-learning based on sequence-processing NNs [14–18], the continual learning problem can also be formulated as a long-span sequence processing task [19]. Corresponding CL sequences can be obtained by concatenating multiple few-shot/meta-learning sub-sequences, where each sub-sequence consists of input/target examples corresponding to the task to be learned in-context. As we'll see in Sec. 3, this setting also allows us to seamlessly express classic desiderata for CL as part of objective functions of the meta-learner.

Once formulated as such a sequence-learning task, we let gradient descent search for CL algorithms achieving the desired CL behaviors in the program space of NN weights. In principle, all typical challenges of CL—such as the stability-plasticity dilemma [20]—are automatically discovered and handled by the gradient-based program search process. Once trained, CL is automated through recursive self-modification dynamics of the trained NN, without requiring any human intervention such as adding extra regularization or setting hyper-parameters for CL. Therefore, we call our method, Automated Continual Learning (ACL).

Our experiments focus on supervised image classification, making use of standard few-shot learning datasets for meta-training, namely, Mini-ImageNet [21, 22], Omniglot [23], and FC100 [24], while we also meta-test on other datasets including MNIST [25], FashionMNIST [26] and CIFAR-10 [27].

**Our core contribution** is a set of focused experiments showing various facets of in-context CL: (1) We first reveal the "in-context catastrophic forgetting" problem using two-task settings (Sec. 4.1) and analyse its emergence (Sec. 4.2). We are not aware of any prior work discussing this problem. (2) We show very promising results of our ACL-trained learning algorithm on the classic Split-MNIST [6, 28] benchmark, outperforming hand-crafted learning algorithms and prior meta-continual learning methods [29–31]. (3) We experimentally illustrate the limitations of ACL on 5-datasets [32] and Split-CIFAR100 by comparing to more recent prompt-based state-of-the-art CL methods [33, 34].

## 2 Background

### 2.1 Continual Learning

The main focus of this work is on continual learning [35, 36] in *supervised* learning settings even though high-level principles we discuss here also transfer to reinforcement learning settings [37]. In addition, we focus on the realm of CL methods that keep model sizes constant (unlike certain CL methods that incrementally add more parameters as more tasks are presented; see, e.g., [38]), and do not make use of any external replay memory (used in other CL methods; see, e.g., [39–43]).

Classic desiderata for a CL system (see, e.g., [44, 45]) are typically summarized as good performance on three metrics: *classification accuracies* on each dataset (their average), *backward transfer* (i.e., impact of learning a new task on the model's performance on previous tasks; e.g., catastrophic forgetting is a negative backward transfer), and *forward transfer* (impact of learning a task for the model's performance on a future task). From a broader perspective of meta-learning systems, we may also measure other effects such as *learning acceleration* (i.e., whether the system leverages previous learning experiences to accelerate future learning); here our primary focus remains the classic CL metrics above.

### 2.2 Few-shot/meta-learning via Sequence Learning

In Sec. 3, we'll formulate continual learning as a long-span sequence processing task. This is a direct extension of the classic few-shot/meta learning formulated as a sequence learning problem. In fact, since the seminal works [14–17] (see also [46]), many sequence processing neural networks (see, e.g., [47–58] including Transformers [59, 18]) have been trained as a meta-learner [13, 12] that learn by observing sequences of training examples (i.e., pairs of inputs and their labels) in-context.

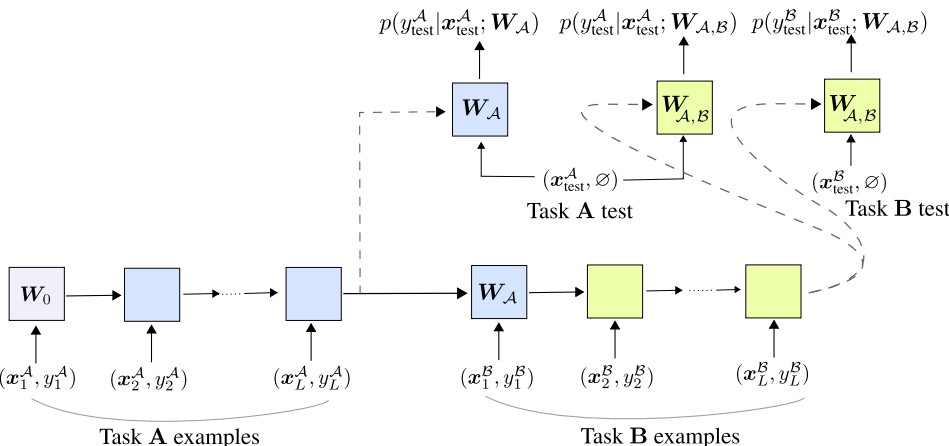

Figure 1: An illustration of meta-training in Automated Continual Learning (ACL) for a self-referential/modifying weight matrix $W_0$. Weights $W_{\mathcal{A}}$ obtained by observing examples for Task A (*blue*) are used to predict a test example for Task A. Weights $W_{\mathcal{A},\mathcal{B}}$ obtained by observing examples for Task A then those for Task B (*yellow*) are used to predict a test example for Task A (backward transfer) as well as a test example for Task B (forward transfer).

Here we briefly review such a formulation. Let $d$, $N$, $K$, $P$ be positive integers. In sequential $N$-way $K$-shot classification settings, a sequence processing NN with a parameter vector $\theta \in \mathbb{R}^P$ observes a pair $(x_t, y_t)$ where $x_t \in \mathbb{R}^d$ is the input and $y_t \in \{1, ..., N\}$ is its label at each step $t \in \{1, ..., N \cdot K\}$, corresponding to $K$ examples for each one of $N$ classes. After the presentation of these $N \cdot K$ examples (often called the *support set*), one extra input $x \in \mathbb{R}^d$ (often called the *query*) is fed to the model without its true label but with an "unknown label" token $\varnothing$ (number of input labels accepted by the model is thus $N + 1$). The model is trained to predict its true label, i.e., the parameters of the model $\theta$ are optimized to maximize the probability $p(y|(x_1, y_1), ..., (x_{N \cdot K}, y_{N \cdot K}), (x, \varnothing); \theta)$ of the correct label $y \in \{1, ..., N\}$ of the input query $x$. Since class-to-label associations are randomized and unique to each sequence $((x_1, y_1), ..., (x_{N \cdot K}, y_{N \cdot K}), (x, \varnothing))$, each such a sequence represents a new (few-shot or meta) learning example to train the model. To be more specific, this is the *synchronous* label setting of Mishra et al. [18] where the learning phase (observing examples, $(x_1, y_1)$ etc.) is separated from the prediction phase (predicting label $y$ given $(x, \varnothing)$). We opt for this variant in our experiments as we empirically find this (at least in our specific settings) more stable than the *delayed* label setting [14] where the model has to make a prediction for every input, and the label is fed to the model with a delay of one time step.

## 2.3 Self-Referential Weight Matrices

Our method (Sec. 3) can be applied to any sequence-processing NN architectures in principle. Nevertheless, certain architectures naturally fit better to parameterize a self-improving continual learner. Here we use the *modern self-referential weight matrix* (SRWM; [19, 60]) to build a generic self-modifying NN. An SRWM is a weight matrix that sequentially modifies itself as a response to a stream of input observations [12, 61]. The modern SRWM belongs to the family of linear Transformers (LTs) a.k.a. Fast Weight Programmers (FWPs; [62–68]). Linear Transformers and FWPs are an important class of the now popular Transformers [59]: unlike the standard ones whose computational requirements grow quadratically and whose state size grows linearly with the context length, LTs/FWPs' complexity is linear and the state size is constant w.r.t. sequence length (like in the standard RNNs). This is an important property for in-context CL, since, conceptually, we want such a CL system to continue to learn for an arbitrarily long, lifelong time span. Moreover, the duality between linear attention and FWPs [67]—and likewise, between linear attention and gradient descent-trained linear layers [69, 70]—have played a key role in certain theoretical analyses of in-context learning capabilities of Transformers [71, 72].

The dynamics of an SRWM [19] are described as follows. Let $d_{\text{in}}$, $d_{\text{out}}$, $t$ be positive integers, and $\otimes$ denote outer product. At each time step $t$, an SRWM $W_{t-1} \in \mathbb{R}^{(d_{\text{out}}+2*d_{\text{in}}+1) \times d_{\text{in}}}$ observes an input

121   $\boldsymbol{x}_t \in \mathbb{R}^{d_{\text{in}}}$, and outputs $\boldsymbol{y}_t \in \mathbb{R}^{d_{\text{out}}}$, while also updating itself to $\boldsymbol{W}_t$ as:

$$[\boldsymbol{y}_t, \boldsymbol{k}_t, \boldsymbol{q}_t, \beta_t] = \boldsymbol{W}_{t-1} \boldsymbol{x}_t \tag{1}$$

$$\boldsymbol{v}_t = \boldsymbol{W}_{t-1} \phi(\boldsymbol{q}_t); \; \bar{\boldsymbol{v}}_t = \boldsymbol{W}_{t-1} \phi(\boldsymbol{k}_t) \tag{2}$$

$$\boldsymbol{W}_t = \boldsymbol{W}_{t-1} + \sigma(\beta_t)(\boldsymbol{v}_t - \bar{\boldsymbol{v}}_t) \otimes \phi(\boldsymbol{k}_t) \tag{3}$$

122   where $\boldsymbol{v}_t, \bar{\boldsymbol{v}}_t \in \mathbb{R}^{(d_{\text{out}}+2*d_{\text{in}}+1)}$ are value vectors, $\boldsymbol{q}_t \in \mathbb{R}^{d_{\text{in}}}$ and $\boldsymbol{k}_t \in \mathbb{R}^{d_{\text{in}}}$ are query and key vectors,
123   and $\sigma(\beta_t) \in \mathbb{R}$ is the learning rate. $\sigma$ and $\phi$ denote sigmoid and softmax functions respectively. $\phi$
124   is typically also applied to $\boldsymbol{x}_t$ in Eq. 1; here we follow Irie et al. [19]'s few-shot image classification
125   setting, and use the variant without it. Eq. 3 corresponds to a rank-one update of the SRWM, from
126   $\boldsymbol{W}_{t-1}$ to $\boldsymbol{W}_t$, through the *delta learning rule* [73, 67] where the self-generated patterns, $\boldsymbol{v}_t, \phi(\boldsymbol{k}_t)$,
127   and $\sigma(\beta_t)$, play the role of *target*, *input*, and *learning rate* of the learning rule respectively. The
128   delta rule is crucial for the performance of LTs [67, 68, 74, 75].

129   The initial weight matrix $\boldsymbol{W}_0$ is the only trainable parameters of this layer, that encodes the initial
130   self-modification algorithm. We use the layer above as a direct replacement to the self-attention layer
131   in the Transformer architecture [59]; and use the multi-head version of the computation above [19].

## 3   Method

133   **Task Formulation.** We formulate continual learning as a long-span sequence learning task. Let
134   $D$, $N$, $K$, $L$ denote positive integers. Consider two $N$-way classification tasks **A** and **B** to be
135   learned sequentially (as we'll see, this can be straightforwardly extended to more tasks). The
136   formulation here applies to both "meta-training" and "meta-test" phases (see Appendix A.1 for more
137   on this terminology). We denote the respective training datasets as $\mathcal{A}$ and $\mathcal{B}$, and test sets as $\mathcal{A}'$
138   and $\mathcal{B}'$. We assume that each datapoint in these datasets consists of one input feature $\boldsymbol{x} \in \mathbb{R}^D$ of
139   dimension $D$ (generically denoted as vector $\boldsymbol{x}$, but it is an image in all our experiments) and one label
140   $y \in \{1, ..., N\}$. We consider two sequences of $L$ training examples $\big((\boldsymbol{x}_1^{\mathcal{A}}, y_1^{\mathcal{A}}), ..., (\boldsymbol{x}_L^{\mathcal{A}}, y_L^{\mathcal{A}})\big)$ and
141   $\big((\boldsymbol{x}_1^{\mathcal{B}}, y_1^{\mathcal{B}}), ..., (\boldsymbol{x}_L^{\mathcal{B}}, y_L^{\mathcal{B}})\big)$ sampled from the respective training sets $\mathcal{A}$ and $\mathcal{B}$. In practice, $L = NK$
142   where $K$ is the number of training examples for each class. By concatenating these two sequences,
143   we obtain one long sequence representing CL examples to be presented as an input sequence to
144   a (left-to-right) auto-regressive model. At the end of the sequence, the model is tasked to make
145   predictions on test examples sampled from both $\mathcal{A}'$ and $\mathcal{B}'$; we assume a single test example for
146   each task (hence, without index): $(\boldsymbol{x}^{\mathcal{A}'}, y^{\mathcal{A}'})$ and $(\boldsymbol{x}^{\mathcal{B}'}, y^{\mathcal{B}'})$ respectively; which we simply denote as
147   $(\boldsymbol{x}_{\text{test}}^{\mathcal{A}}, y_{\text{test}}^{\mathcal{A}})$ and $(\boldsymbol{x}_{\text{test}}^{\mathcal{B}}, y_{\text{test}}^{\mathcal{B}})$ instead.

148   Our model is a self-referential NN that modifies its own weight matrices as a function of input
149   observations. To simplify the notation, we denote the *state* of our self-referential NN as a
150   *single* SRWM $\boldsymbol{W}_*$ (even though it may have many of them in practice) where we'll replace $*$
151   by various symbols representing the context/inputs it has observed. Given a training sequence
152   $\big((\boldsymbol{x}_1^{\mathcal{A}}, y_1^{\mathcal{A}}), ..., (\boldsymbol{x}_L^{\mathcal{A}}, y_L^{\mathcal{A}}), (\boldsymbol{x}_1^{\mathcal{B}}, y_1^{\mathcal{B}}), ..., (\boldsymbol{x}_L^{\mathcal{B}}, y_L^{\mathcal{B}})\big)$, our model auto-regressively consumes one input
153   at a time, from left to right, in the auto-regressive fashion. Let $\boldsymbol{W}_{\mathcal{A}}$ denote the state of the SRWM that
154   has consumed the first part of the sequence, i.e., the examples from Task **A**, $(\boldsymbol{x}_1^{\mathcal{A}}, y_1^{\mathcal{A}}), ..., (\boldsymbol{x}_L^{\mathcal{A}}, y_L^{\mathcal{A}})$,
155   and let $\boldsymbol{W}_{\mathcal{A},\mathcal{B}}$ denote the state of our SRWM having observed the entire sequence.

156   **ACL Meta-Training Objectives.** The ACL meta-training objective function tasks the model to
157   correctly predict the test examples of all tasks learned so far at each task boundaries. That is, in the
158   case of two-task scenario described above (learning Task **A** then Task **B**), we use the weight matrix
159   $\boldsymbol{W}_{\mathcal{A}}$ to predict the label $y_{\text{test}}^{\mathcal{A}}$ from input $(\boldsymbol{x}_{\text{test}}^{\mathcal{A}}, \varnothing)$, and we use the weight matrix $\boldsymbol{W}_{\mathcal{A},\mathcal{B}}$ to predict the
160   label $y_{\text{test}}^{\mathcal{B}}$ from input $(\boldsymbol{x}_{\text{test}}^{\mathcal{B}}, \varnothing)$ *as well as* the label $y_{\text{test}}^{\mathcal{A}}$ from input $(\boldsymbol{x}_{\text{test}}^{\mathcal{A}}, \varnothing)$. By letting $p(y|\boldsymbol{x}; \boldsymbol{W}_*)$
161   denote the model's output probability for label $y \in \{1, .., N\}$ given input $\boldsymbol{x}$ and model weights/state
162   $\boldsymbol{W}_*$, the ACL objective can be expressed as:

$$\underset{\theta}{\text{minimize}} - \big(\log(p(y_{\text{test}}^{\mathcal{A}}|\boldsymbol{x}_{\text{test}}^{\mathcal{A}}; \boldsymbol{W}_{\mathcal{A}})) + \log(p(y_{\text{test}}^{\mathcal{B}}|\boldsymbol{x}_{\text{test}}^{\mathcal{B}}; \boldsymbol{W}_{\mathcal{A},\mathcal{B}})) + \log(p(y_{\text{test}}^{\mathcal{A}}|\boldsymbol{x}_{\text{test}}^{\mathcal{A}}; \boldsymbol{W}_{\mathcal{A},\mathcal{B}}))\big) \tag{4}$$

163   for an arbitrary input meta-training sequence $\big((\boldsymbol{x}_1^{\mathcal{A}}, y_1^{\mathcal{A}}), ..., (\boldsymbol{x}_L^{\mathcal{A}}, y_L^{\mathcal{A}}), (\boldsymbol{x}_1^{\mathcal{B}}, y_1^{\mathcal{B}}), ..., (\boldsymbol{x}_L^{\mathcal{B}}, y_L^{\mathcal{B}})\big)$
164   (which is extensible to mini-batches with multiple such sequences), where $\theta$ denotes the model
165   parameters (for the SRWM layer, it is the initial weights $\boldsymbol{W}_0$). Figure 1 illustrates the overall
166   meta-training process of ACL.

Table 1: 5-way classification accuracies using 15 (meta-test training) examples for each class in the context. Each row is a single model. **Bold** numbers highlight cases where in-context catastrophic forgetting is avoided through ACL.

| | | | Meta-Test Tasks: Context/Train (top) & Test (bottom) | | | | | |
| --- | --- | --- | --- | --- | --- | --- | --- | --- |
| Meta-Training Tasks | | | A | A $\to$ B | | B | B $\to$ A | |
| Task A | Task B | ACL | A | B | A | B | A | B |
| Omniglot | Mini-ImageNet | No | 97.6 ± 0.2 | 52.8 ± 0.7 | 22.9 ± 0.7 | 52.1 ± 0.8 | 97.8 ± 0.3 | 20.4 ± 0.6 |
| | | Yes | 98.3 ± 0.2 | 54.4 ± 0.8 | **98.2** ± 0.2 | 54.8 ± 0.9 | 98.0 ± 0.3 | **54.6** ± 1.0 |
| FC100 | Mini-ImageNet | No | 49.7 ± 0.7 | 55.0 ± 1.0 | 21.3 ± 0.7 | 55.1 ± 0.6 | 49.9 ± 0.8 | 21.7 ± 0.8 |
| | | Yes | 53.8 ± 1.7 | 52.5 ± 1.2 | **46.2** ± 1.3 | 59.9 ± 0.7 | 45.5 ± 0.9 | **53.0** ± 0.6 |

Table 2: Similar to Table 1 above but using MNIST and CIFAR-10 (unseen domains) for meta-testing.

| | | | Meta-Test Tasks: Context/Train (top) & Test (bottom) | | | | | |
| --- | --- | --- | --- | --- | --- | --- | --- | --- |
| Meta-Training Tasks | | | MNIST | MNIST $\to$ CIFAR-10 | | CIFAR-10 | CIFAR-10 $\to$ MNIST | |
| Task A | Task B | ACL | MNIST | CIFAR-10 | MNIST | CIFAR-10 | MNIST | CIFAR-10 |
| Omniglot | Mini-ImageNet | No | 71.1 ± 4.0 | 49.4 ± 2.4 | 43.7 ± 2.3 | 51.5 ± 1.4 | 68.9 ± 4.1 | 24.9 ± 3.2 |
| | | Yes | 75.4 ± 3.0 | 50.8 ± 1.3 | **81.5** ± 2.7 | 51.6 ± 1.3 | 77.9 ± 2.3 | **51.8** ± 2.0 |
| FC100 | Mini-ImageNet | No | 60.1 ± 2.0 | 56.1 ± 2.3 | 17.2 ± 3.5 | 54.4 ± 1.7 | 58.6 ± 1.6 | 21.2 ± 3.1 |
| | | Yes | 70.0 ± 2.4 | 51.0 ± 1.0 | **68.2** ± 2.7 | 59.2 ± 1.7 | 66.9 ± 3.4 | **52.5** ± 1.3 |

The ACL objective function above (Eq. 4) is simple but encapsulates desiderata for continual learning (Sec. 2.1). The last term of Eq. 4 with $p(y_{\text{test}}^{\mathcal{A}}|\boldsymbol{x}_{\text{test}}^{\mathcal{A}}; \boldsymbol{W}_{\mathcal{A},\mathcal{B}})$ or schematically $p(\mathcal{A}'|\mathcal{A}, \mathcal{B})$, optimizes for *backward transfer*: (1) remembering the first task **A** after learning **B** (combatting catastrophic forgetting), and (2) leveraging learning of **B** to improve performance on the past task **A**. The second term of Eq. 4, $p(y_{\text{test}}^{\mathcal{B}}|\boldsymbol{x}_{\text{test}}^{\mathcal{B}}; \boldsymbol{W}_{\mathcal{A},\mathcal{B}})$ or schematically $p(\mathcal{B}'|\mathcal{A}, \mathcal{B})$, optimizes *forward transfer* leveraging the past learning experience of **A** to improve predictions in the second task **B**, in addition to simply learning to solve Task **B** from the corresponding training examples. To complete, the first term of Eq. 4 is the single-task meta-learning objective for Task **A**.

**Overall Model Architecture.** As we mention in Sec. 2, in our NN architecture, the core sequential dynamics of CL are learned by the self-referential layers. However, as an image-processing NN, our model makes use of a vision backend. We use the "Conv-4" architecture [21] (typically used in the context of few-shot learning) in all our experiments, except in the last one where we use a pre-trained vision Transformer [76]. Overall, the model takes an image as input, process it through a feedforward vision NN, whose output is fed to the SRWM-layer block. Note that this is one of the limitations of this work: more general ACL should also learn to modify the vision components.[2]

Another crucial architectural choice that is specific to continual/multi-task image processing is normalization layers (see also Bronskill et al. [78]). Typical NNs used in few-shot learning (e.g., Vinyals et al. [21]) contain batch normalization (BN; [79]) layers. All our models use instance normalization (IN; [80]) instead of BN because in our preliminary experiments, we expectably found IN to generalize much better than BN layers in the CL setting.

## 4  Experiments

### 4.1  Two-Task Setting: Comprehensible Study

We first reveal the problem of "in-context catastrophic forgetting" and show how our ACL method (Sec. 3) can overcome it. As a minimum setting for this, we focus on the two-task "domain-

---

[2]One "straightforward" architecture fitting the bill is an MLP-mixer architecture (Tolstikhin et al. [77]; built of several linear layers), where all linear layers are replaced by the self-referential linear layers of Sec. 2.3. While we implemented such a model, it turned out to be too slow for us to conduct corresponding experiments. Our public code will include a "self-referential MLP-mixer" implementation, but for further experiments, we leave the future work on such an architecture using more efficient CUDA kernels.

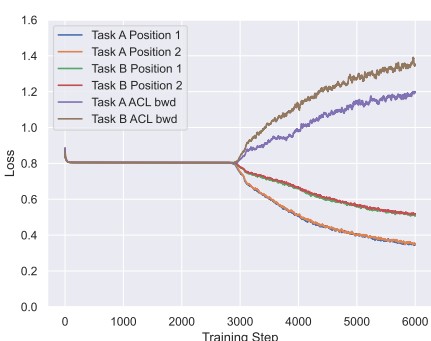 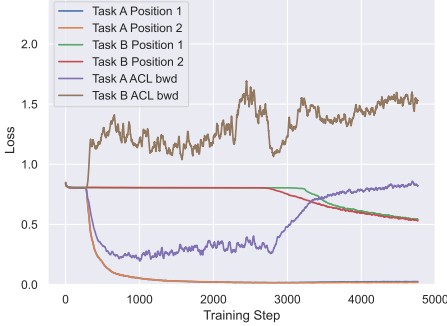

(a) Case: Two tasks are learned simultaneously.     (b) Case: One task is learned first (here Task A).

Figure 2: **ACL/No**-case meta-training curves displaying 6 individual meta-training loss terms, when the last term of the ACL objective (the backward tranfer loss; "*Task A ACL bwd*" and "*Task B ACL bwd*" in the legend) is **not** minimized (**ACL/No** case in Tables 1 and 2). Here Task A is Omniglot and Task B is Mini-ImageNet. We observe that, in both cases, without explicit minimization, backward transfer capability (*purple* and *brown* curves) of the learned learning algorithm gradually degrades as it learns to learn a new task (all other colors), causing in-context catastrophic forgetting. Note that *blue/orange* and *green/red* curve pairs almost overlap; indicating that when a task is learned, the model can learn it whether it is in the first or second segment of the continual learning sequence.

incremental" CL setting (see Appendix A.1). We consider two meta-training task combinations: Omniglot [23] and Mini-ImageNet [21, 22] or FC100 [24] (which is based on CIFAR100 [27]) and Mini-ImageNet. The order of appearance of two tasks within meta-training sequences is alternated for every batch. Appendix A.2 provides further details. We compare systems trained with or without the backward transfer term in the ACL loss (the last term in Eq. 4).

Unless otherwise indicated (e.g, later for classic Split-MNIST; Sec. 4.3), all tasks are configured to be a 5-way classification task. This is one of the classic configurations for few-shot learning tasks, and also allows us to evaluate the principle of ACL with reasonable computational costs (like any sequence learning-based meta-learning methods, scaling this to many more classes is challenging; we also discuss this in Sec. 5). For standard datasets such as MNIST, we split the dataset into sub-datasets of disjoint classes [81]: for example for MNIST which is originally a 10-way classification task, we split it into two 5-way tasks, one consisting of images of class '0' to '4' ('MNIST-04'), and another one made of class '5' to '9' images ('MNIST-59'). When we refer to a dataset without specifying the class range, we refer to the first sub-set. Unless stated otherwise, we concatenate 15 examples from each class for each task in the context for both meta-training and meta-testing (resulting in sequences of length 75 for each task). All images are resized to $32 \times 32$-size 3-channel images, and normalized according to the original dataset statistics. We refer to Appendix A for further details.

Table 1 shows the results when the models are meta-tested on the test sets of the corresponding few-shot learning datasets used for meta-training. We observe that for both pairs of meta-training tasks, the models without the ACL loss *catastrophically forget* the first task after learning the second one: the accuracy on the first task is at the chance level of about 20% for 5-way classification after learning the second task in-context (see rows with "ACL No"). The ACL loss clearly addresses this problem: the ACL-learned CL algorithms preserve the performance of the first task. This effect is particularly pronounced in the Omniglot/Mini-ImageNet case (involving two very different domains).

Table 2 shows evaluations of the same models but using two standard datasets, 5-way MNIST and CIFAR-10, for meta-testing. Again, ACL-trained models better preserve the memory of the first task after learning the second one. In the Omniglot/Mini-ImageNet case, we even observe certain positive backward tranfer effects: in particular, in the "MNIST-then-CIFAR10" continual learning case, the performance on MNIST noticeably improves after learning CIFAR10 (possibly leveraging 'more data' provided in-context).

## 4.2 Analysis: Emergence of In-Context Catastrophic Forgetting

Now we closely look at the emergence of "in-context catastrophic forgetting" during meta-training for the baseline models trained **without** the backward transfer term (the last/third term in Eq. 4) in

Table 3: Classification accuracies (%) on the **Split-MNIST** domain-incremental (DIL) and class-incremental learning (CIL) settings [6]. Both tasks are 5-task CL problems. For the CIL case, we also report the 2-task case for which we can directly evaluate our out-of-the-box ACL meta-learner of Sec. 4.1 (trained with a 5-way output and the 2-task ACL loss) which, however, is not applicable (N.A.) to the 5-task CIL requiring a 10-way output. Mean/std over 10 training/meta-testing runs. **No method here requires replay memory**. See Appendix A.7 & B for further details and discussions.

| Method | Domain Incremental | Class Incremental | |
| | 5-task | 2-task | 5-task |
| --- | --- | --- | --- |
| Plain Stochastic Gradient Descent (SGD) | $63.2 \pm 0.4$ | $48.8 \pm 0.1$ | $19.5 \pm 0.1$ |
| Adam | $55.2 \pm 1.4$ | $49.7 \pm 0.1$ | $19.7 \pm 0.1$ |
| Adam + L2 | $66.0 \pm 3.7$ | $51.8 \pm 1.9$ | $22.5 \pm 1.1$ |
| Elastic Weight Consolidation (EWC) | $58.9 \pm 2.6$ | $49.7 \pm 0.1$ | $19.8 \pm 0.1$ |
| Online EWC | $57.3 \pm 1.4$ | $49.7 \pm 0.1$ | $19.8 \pm 0.1$ |
| Synaptic Intelligence (SI) | $64.8 \pm 3.1$ | $49.4 \pm 0.2$ | $19.7 \pm 0.1$ |
| Memory Aware Synapses (MAS) | $68.6 \pm 6.9$ | $49.6 \pm 0.1$ | $19.5 \pm 0.3$ |
| Learning w/o Forgetting (LwF) | $71.0 \pm 1.3$ | - | $24.2 \pm 0.3$ |
| Online-aware Meta Learning (OML) | $69.9 \pm 2.8$ | $46.6 \pm 7.2$ | $24.9 \pm 4.1$ |
| + optimized # meta-testing iterations | $73.6 \pm 5.3$ | $62.1 \pm 7.9$ | $34.2 \pm 4.6$ |
| Generative Meta-Continual Learning (GeMCL) | $63.8 \pm 3.8$ | $91.2 \pm 2.8$ | $79.0 \pm 2.1$ |
| ACL (Out-of-the-box, DIL, 2-task ACL model; Sec. 4.1) | $72.2 \pm 0.9$ | $71.5 \pm 5.9$ | N.A. |
| + meta-finetuned with 5-task ACL loss, Omniglot | $\mathbf{84.5} \pm 1.6$ | $\mathbf{96.0} \pm 1.0$ | $\mathbf{84.3} \pm 1.2$ |

the ACL objective loss (corresponding to the **ACL/No** cases in Tables 1 and 2). We focus on the Omniglot/Mini-ImageNet case, but similar trends can also be observed in the FC100/Mini-ImageNet case. Figures 2a and 2b show two representative cases we typically observe. These figures show an evolution of six individual meta-training loss terms (the lower the better), reported separately for the cases where Task A (here Omniglot) or Task B (here Mini-ImageNet) appears at the first (1) or second (2) position in the 2-task CL meta-training training sequences. 4 out of 6 curves correspond to the learning progress, showing whether the model becomes capable of in-context learning the given task (A or B) at the given position (1 or 2). The 2 remaining curves are the ACL backward tranfer losses, also measured for Task A and B separately here.

Figure 2a shows the case where two tasks are learned about at the same time. We observe that when the learning curves go down, the ACL losses go up, indicating that more the model learns, more it tends to forget the task in-context learned previously. We also find this same trend when one task is learned before the other one as is the case in Figure 2b. Here Task A alone is learned first; while Task B is not learned, both learning and ACL curves go down for Task A (essentially, as the model does not learn the second task, there is no force that encourages forgetting). After around 3000 steps, the model also starts learning Task B. From this point, the ACL loss for Task A also starts to go up, indicating again an *opposing force effect* between learning a new task and remembering a past task. These observations clearly indicate that, without explicitly taking into account the backward transfer loss as part of learning objectives, our gradient descent search tends to find solutions/CL algorithms that prefer to erase previously learned knowledge (this is rather intuitive; it seems easier to find such algorithms that ignore any influence of the current learning to past learning than those that also preserve prior knowledge). In all cases, we find our ACL objective to be crucial for the learned CL algorithms to be capable of remembering the old task while also learning the new one.

## 4.3 General Evaluation

**Evaluation on Standard Split-MNIST.** Here we evaluate ACL on the standard Split-MNIST task in domain-incremental and class-incremental settings [6, 28], and compare its performance to existing CL and meta-CL algorithms (see Appendix A.7 for full references of these methods). Our comparison focuses on methods that do not require replay memory. Table 3 shows the results. Since our ACL-trained models are general-purpose learners, they can be directly evaluated (meta-tested) on a new task, here Split-MNIST. The second-to-last row of Table 3, "ACL (Out-of-the-box model)", corresponds to our model from Sec. 4.1 meta-trained on Omniglot and Mini-ImageNet using the 2-task ACL objective. It performs very competitively with the best existing methods in the domain-

incremental setting, while it largely outperforms them (all but another meta-CL method, GeMCL) in the 2-task class-incremental setting. The same model can be further meta-finetuned using the 5-task version of the ACL loss (here we only used Omniglot as the meta-training data). The resulting model (the last row of Table 3) outperforms all other methods in all settings studied here. Note that on the 'in-domain' Omniglot test set, ACL and GeMCL perform similarly (see Appendix B.2/Table 9). We are not aware of any existing hand-crafted CL algorithms that can achieve ACL's performance without any replay memory. We refer to Appendix A.7/B for further discussions and ablation studies.

**Evaluation on diverse task domains.** Using the setting of Sec. 4.1, we also evaluate our ACL-trained models for CL involving more tasks/domains; using meta-test sequences made of MNIST, CIFAR-10, and Fashion MNIST. We also evaluate the impact of the number of tasks in the ACL objective: in addition to the model meta-trained on Omniglot/Mini-ImageNet (Sec. 4.1), we also meta-train a model (with the same architecture and hyper-parameters) using 3 tasks, Omniglot, Mini-ImageNet, and FC100, using the 3-task ACL objective (see Appendix A.5); which is meta-trained not only on longer CL sequences but also on more data. The full results of this experiment can be found in Appendix B.4. We find that the two ACL-trained models are indeed capable of retaining the knowledge without catastrophic forgetting for multiple tasks during meta-testing, while the performance on prior tasks gradually degrades as the model learns new tasks, and performance on new tasks becomes moderate (see also Sec. 5 on limitations). The 3-task version outperforms the 2-task one overall, encouragingly indicating a potential for further improvements even with a fixed parameter count.

**Going beyond: limitations and outlook.** The experiments presented above effectively demonstrate the possibility to encode a continual learning algorithm into self-referential weight matrices, that outperforms handcrafted learning algorithms and existing metalearning approaches for CL. While we consider this as an important result for metalearning and in-context learning in general, we note that current state-of-the-art CL methods use neither regularization-based CL algorithms nor meta-continual learning methods we mention above, but the so-called *learning to prompt* (L2P)-family of methods [33, 34] that leverage pre-trained models, namely a vision Transformer (ViT) pre-trained on ImageNet [76]. A natural question we should ask is whether we could foresee ACL beyond the scope considered so far, and evaluate it in such a setting. To study this, we take a pre-trained (frozen) vision model, and add self-referential layers (to be meta-trained from scratch) on top of it to build a continual learner. This allows us to highlight an important challenge of in-context CL in what follows.

We use two tasks from the L2P works above [33, 34]: 5-datasets [32] and Split-CIFAR-100, in the class-incremental setting, but we focus on a "*mini*" versions thereof: we only use the two first classes within each task (i.e., *2-way* version) and for Split-CIFAR100, we only use the 5 first tasks; as we'll see, this setting is enough to illustrate an important limitation of in-context CL. Again following L2P [33, 34], we use ViT-B/16 [76] (available via PyTorch) as the pre-trained vision model, which we keep frozen. We use the same configuration for the self-referential component from the Split-MNIST experiment. We meta-train the resulting model using Mini-ImageNet and Omniglot with the 5-task ACL loss. Table 4 shows the results. Even in this simple "mini" version of the tasks, ACL's performance is far behind that of L2P methods. Notably, the frozen ImageNet-pre-trained features with the meta-learner trained on Mini-ImageNet and Omniglot are not enough to perform well on the 5-th task of Split-CIFAR100, and SVHN and notMNIST of 5-datasets. This shows the necessity to meta-train on more diverse tasks for in-context CL to be possibly successful in more general settings.

Table 4: Experiments with "*mini*" Split-CIFAR100 and 5-datasets tasks. Meta-training is done using **Mini-ImageNet** and **Omniglot**. All meta-evaluation images are therefore from unseen domains. Numbers marked with * are *reference* numbers (evaluated in the more challenging, original version of these tasks) which can not be directly compared to ours.

| | Split-CIFAR100 | | 5-datasets | |
|---|---|---|---|---|
| L2P [34] | *83.9* $\pm$ 0.3 | | *81.1* $\pm$ 0.9 | |
| DualPrompt [34] | *86.5* $\pm$ 0.3 | | *88.1* $\pm$ 0.4 | |
| ACL (Individual Task) | Task 1 | 95.9 $\pm$ 0.9 | CIFAR10 | 91.3 $\pm$ 1.2 |
| | Task 2 | 85.6 $\pm$ 3.6 | MNIST | 98.9 $\pm$ 0.3 |
| | Task 3 | 93.4 $\pm$ 1.4 | Fashion | 93.5 $\pm$ 2.0 |
| | Task 4 | 97.0 $\pm$ 0.7 | SVHN | 66.1 $\pm$ 9.4 |
| | Task 5 | 67.6 $\pm$ 7.0 | notMNIST | 76.3 $\pm$ 6.7 |
| ACL | 68.3 $\pm$ 2.0 | | 61.5 $\pm$ 2.1 | |

## 5  Discussion

**Other Limitations.** In addition to the limitations already mentioned above, here we discuss others. First of all, as an in-context/learned learning algorithm, there are challenges in terms of both domain and length generalization (we qualitatively observe these to some extent in Sec. 4; further discussion and experimental results are presented in Appendix B.3 & B.5). Regarding the length generalization, we note that unlike the standard "quadratic" Transformers, linear Transformers/FWPs-like SRWMs can be trained by *carrying over states* across two consecutive batches for arbitrarily long sequences. Such an approach has been successfully applied to language modeling with FWPs [67]. This possibility, however, has not been investigated here, and is left for future work. Also, directly scaling ACL for real-world tasks requiring many more classes does not seem straightforward: it would require very long training sequences. That said, it may be possible that ACL could be achieved without exactly following the process we propose; as we discuss below for the case of LLMs, certain real-world data may naturally give rise to an ACL-like objective. This work is also limited to the task of image classification, which can be solved by feedforward NNs. Future work may investigate the possibility to extend ACL to continual learning of sequence learning tasks, such as continually learning new languages. Finally, ACL learns CL algorithms that are specific to the pre-specified model architecture; more general meta-learning algorithms may aim at achieving learning algorithms that are applicable to any model, as is the case for many classic learning algorithms.

**Related work.** There are several recent works that are catagorized as 'meta-continual learning' or 'continual meta-learning' (see, e.g., [29, 30, 82–84, 51]). For example, Javed and White [29], Beaulieu et al. [30] use "model-agnostic meta-learning" (MAML; [85, 86]) to meta-learn *representations* for CL while still making use of classic learning algorithms for CL; this requires tuning of the learning rate and number of iterations for optimal performance during CL at meta-test time (see, e.g., Appendix A.7). In contrast, our approach learn *learning algorithms* in the spirit of Hochreiter et al. [14], Younger et al. [15]; this may be categorized as 'in-context continual learning.' Several recent works (see, e.g., [87, 88]) mention the possibility of such in-context CL but existing works [19, 89, 90] that learn multiple tasks sequentially in-context do not focus on catastrophic forgetting which is one of the central challenges of CL. Here we show that in-context learning also suffers from catastrophic forgetting in general (Sec. 4.1-4.2) and propose ACL to address this problem. We also note that the use of SRWM is relevant to 'continual meta-learning' since with a regular sequence processor with slow weights, there remains the question of how to continually learn the slow weights (meta-parameters). In principle, recursive self-modification as in SRWM is an answer to this question as it collapses such meta-levels into single self-reference [12]. We also refer to [91–93] for other prior work on meta-continual learning.

**Artificial v. Natural ACL in Large Language Models?**  Recently, "on-the-fly" few-shot/meta learning capability of sequence processing NNs has attracted broader interests in the context of large language models (LLMs; [94]). In fact, the task of language modeling itself has a form of *sequence processing with error feedback* (essential for meta-learning [95]): the correct label to be predicted is fed to the model with a delay of one time step in an auto-regressive manner. Trained on a large amount of text covering a wide variety of credit assignment paths, LLMs exhibit certain sequential few-shot learning capabilities in practice [96]. This was rebranded as *in-context learning*, and has been the subject of numerous recent studies (e.g., [97–103, 71, 72]). Here we explicitly/artificially construct ACL meta-training sequences and objectives, but in modern LLMs trained on a large amount of data mixing a large diversity of dependencies using a large backpropagation span, it is conceivable that some ACL-like objectives may naturally appear in the data.

## 6  Conclusion

Our Automated Continual Learning (ACL) trains sequence-processing self-referential neural networks (SRNNs) to learn their own in-context continual (meta-)learning algorithms. ACL encodes classic desiderata for continual learning (e.g., forward and backward transfer) into the objective function of the meta-learner. ACL uses gradient descent to deal with classic challenges of CL, to automatically discover CL algorithms with good behavior. Once trained, our SRNNs autonomously run their own CL algorithms without requiring any human intervention. Our experiments reveal the original problem of in-context catastrophic forgetting, and demonstrate the effectiveness of the proposed approach to combat it. We demonstrate very promising results on the classic Split-MNIST benchmark where existing hand-crafted algorithms fail, while also discussing its limitations in more general scenarios. We believe this comprehensive study to be an important step for in-context CL research.

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

# A  Experimental Details

## A.1  Continual and Meta-learning Terminologies

We review the following classic terminologies of continual learning and meta-learning used throughout this paper.

**Continual learning.**  "Domain-incremental learning (DIL)" and "class-incremental learning (CIL)" are two classic settings in continual learning [104, 28, 6]. They differ as follows. Let $M$ and $N$ denote positive integers. Consider continual learning of $M$ tasks where each task is an $N$-way classification. In the DIL case, a model has an $N$-way output classification layer, i.e., the class '0' of the first task shares the same weights as the class '0' of the second task, and so on. In the CIL case, a model's output dimension is $N * M$; the class indices of different tasks are not shared, neither are the corresponding weights in the output layer. In our experiments, all CIL models have the $(N * M)$-way output from the first task (instead of progressively increasing the output size). In this work, we skip the third variant called "task-incremental learning" which assumes that we have access to the task identity as an extra input, as it makes the CL problem almost trivial. CIL is typically reported to be the hardest setting among them.

**Meta-learning.**  We need to introduce "meta-training" and "meta-test" terminologie since each of these phases involve "training/test" processes within itself. Each of them requires the corresponding training and test examples. We refer to these as "meta-training training/test examples", and "meta-test training/test examples" following the terminology of Beaulieu et al. [30]. While these are rather "heavy" terminologies, they are unambiguous and help avoid potential confusions. In both phases, our sequence-processing neural net observes a sequence of (meta-training or meta-test) training examples—each consisting of input features and a correct label—, and the resulting states of the sequence processor (i.e., weights in the case of SRWM) are used to make predictions on (meta-training or meta-test) test examples—input features presented to the model without its label. During the meta-training phase, we modify the trainable parameters of the meta-learner through gradient descent minimizing the meta-learning loss function (using backpropagation through time). During meta-testing, no human-designed optimization for weight modification is used anymore; the SRWMs modify their own weights following their own learning rules defined as their forward pass (Eqs. 1-3). In connection with the now-popular in-context learning [96], we also refer to a (meta-training or meta-test) training-example sequence as *context*.

## A.2  Datasets

For classic image classification datasets such as MNIST [25], CIFAR10 [27], and FashionMNIST (FMNIST; Xiao et al. [26]) we refer to the original references for details.

For Omniglot [23], we use Vinyals et al. [21]'s 1028/172/432-split for the train/validation/test set, as well as their data augmentation methods using rotation of 90, 180, and 270 degrees. Original images are grayscale hand-written characters from 50 different alphabets. There are 1632 different classes with 20 examples for each class.

Mini-ImageNet contains color images from 100 classes with 600 examples for each class. We use the standard train/valid/test class splits of 64/16/20 following [22].

FC100 is based on CIFAR100 [27]. 100 color image classes (600 images per class, each of size $32 \times 32$) are split into train/valid/test classes of 60/20/20 [24].

The "5-datasets" dataset [32] consists of 5 datasets: CIFAR10, MNIST, FashionMNST, SVNH [105], and notMNIST [106].

Split-CIFAR100 is also based on CIFAR100. The standard setting splits CIFAR100 into 10 10-way classification tasks.

Meta-train/test sequence construction procedure. We use `torchmeta` [107] which provides common few-shot/meta learning settings for these datasets to sample and construct their meta-train/test datasets. The construction of "meta-training training" sequences for an $N$-way classification, using a dataset containing $C$ classes works as follows; for each sequence, we sample $N$ random but distinct classes out of $C$ ($N < C$). The resulting classes are re-labelled such that each class is assigned to one out of $N$ distinct random label index which is unique to the sequence. For each of these $N$ classes, we sample $K$ examples. We randomly order these $N * K$ examples to obtain a sequence. Each such a sequence "simulates" an unknown task the model has to learn.

## A.3 Training Details & Hyper-Parameters

We use the same model and training hyper-parameters in all our experiments. All hyper-parameters are summarized in Table 5. We use the Adam optimizer with the standard Transformer learning rate warmup scheduling [59]. The vision backend is the classic 4-layer convolutional NN of Vinyals et al. [21]. Most configurations follow those of Irie et al. [19]; except that we initialize the 'query' sub-matrix in the self-referential weight matrix using a normal distribution with a mean value of 0 and standard deviation of $0.01/\sqrt{d_{\text{head}}}$ while other sub-matrices use an std of $1/\sqrt{d_{\text{head}}}$ (motivated by the fact that a generated query vector is immediately multiplied with the same SRWM to produce a value vector). For any further details, we'll refer the readers to our public code we'll release upon acceptance. We conduct our experiments using a single V100-32GB, 2080-12GB or P100-16GB GPUs, and the longest single training run takes about one day.

Table 5: Hyper-parameters.

| Parameters | Values |
|---|---|
| Number of SRWM layers | 2 |
| Total hidden size | 256 |
| Feedforward block multiplier | 2 |
| Number of heads | 16 |
| Batch size | 16 or 32 |

## A.4 Evaluation Procedure

For evaluation on few-shot learning datasets (i.e., Omniglot, Mini-Imagenet and FC100), we use 5 different sets consisting of 32 K random test episodes each, and report mean and standard deviation.

For evaluation on standard datasets, we use 5 different random support sets for in-context learning, and evaluate on the entire test set. We report the corresponding mean and standard deviation across these 5 evaluation runs.

For the Split-MNIST experiment, we do 10 meta-testing runs to compute the mean and standard deviation as the baseline models are also trained for 10 runs in Hsu et al. [6] (see other details in Appendix A.7).

## A.5 ACL Objectives with More Tasks

We can straightforwardly extend the 2-task version of ACL presented in Sec. 3 to more tasks. In the 3-task case (we denote the three tasks as **A**, **B**, and **C**) used in Sec. 4.3, the objective function contains six terms. Following three terms are added to Eq. 4:

$$- \Big( \log(p(y_{\text{test}}^{\mathcal{C}}|\boldsymbol{x}_{\text{test}}^{\mathcal{C}}; \boldsymbol{W}_{\mathcal{A},\mathcal{B},\mathcal{C}})) + \log(p(y_{\text{test}}^{\mathcal{B}}|\boldsymbol{x}_{\text{test}}^{\mathcal{B}}; \boldsymbol{W}_{\mathcal{A},\mathcal{B},\mathcal{C}})) + \log(p(y_{\text{test}}^{\mathcal{A}}|\boldsymbol{x}_{\text{test}}^{\mathcal{A}}; \boldsymbol{W}_{\mathcal{A},\mathcal{B},\mathcal{C}})) \Big)$$

This also naturally extends to the 5-task loss used in the Split-MNIST experiment (Table 3). As one can observe, the number of terms rapidly/quadratically increases with the number of tasks. Nevertheless, computing these loss terms isn't immediately impractical because they essentially just require forwarding the network for one step, for many independent inputs/images. This can be heavily parallelized as a batch operation. While this can be a concern when scaling up more, a natural open research question is whether we really need all these terms in the case we have many more tasks.

Table 6: Impact of the choice of meta-validation datasets. Classification accuracies (%) on three datasets: **Split-CIFAR-10**, **Split-Fashion MNIST** (Split-FMNIST), and **Split-MNIST** in the **domain-incremental** setting (we omit "Split-" in the second column). "OOB" denotes "out-of-the-box". "mImageNet" here refers to mini-ImageNet.

| Meta-Finetune Datasets | Meta-Validation Sets | Meta-Test on Split- | | |
| --- | --- | --- | --- | --- |
| | | MNIST | FMNIST | CIFAR-10 |
| None (OOB: 2-task ACL; Sec. 4.1) | Omniglot + mImageNet | $72.2 \pm 0.9$ | $75.6 \pm 0.7$ | $65.3 \pm 1.6$ |
| Omniglot | MNIST | $\mathbf{84.3} \pm 1.2$ | $78.1 \pm 1.9$ | $55.8 \pm 1.2$ |
| | FMNIST | $81.6 \pm 1.3$ | $\mathbf{90.4} \pm 0.5$ | $59.5 \pm 2.1$ |
| | CIFAR10 | $75.2 \pm 2.3$ | $78.2 \pm 0.9$ | $\mathbf{63.4} \pm 1.4$ |
| Omniglot + mImageNet | MNIST | $\mathbf{76.6} \pm 1.4$ | $85.3 \pm 1.1$ | $66.2 \pm 1.1$ |
| | FMNIST | $73.2 \pm 2.3$ | $\mathbf{89.9} \pm 0.6$ | $66.6 \pm 0.7$ |
| | CIFAR10 | $76.3 \pm 3.0$ | $88.1 \pm 1.3$ | $\mathbf{68.6} \pm 0.5$ |

Ideally, we want these models to 'systematically generalize' to more tasks even when they are trained with only a handful of them [108]. This is an interesting research question on generalization to be studied in a future work.

## A.6  Auxiliary 1-shot Learning Objective

In practice, instead of training the models only for "15-shot learning," we also add an auxiliary loss for 1-shot learning. This naturally encourages the models to learn in-context from the first examples.

## A.7  Details of the Split-MNIST experiment

Here we provide details of the Split-MNIST experiments presented in Sec. 4 and Table 3.

Split-MNIST is obtained by transforming the classic 10-class single-task MNIST dataset into a sequence of 5 tasks by partitioning the 10 classes into 5 groups/pairs of two classes each, in a fixed order from 0 to 9 (i.e., grouping 0/1, 2/3, 4/5, 6/7, and 8/9). Regarding the difference between domain/class-incremental settings, we refer to Appendix A.1.

The baseline methods presented in Table 3 include: standard SGD and Adam optimizers, Adam with the L2 regularization, elastic weight consolidation [9] and its online variant [10], synaptic intelligence [11], memory aware synapses [109], learning without forgetting (LwF; Li and Hoiem [110]). For these methods, we directly take the numbers reported in Hsu et al. [6] for the 5-task domain/class-incremental settings.

For the 2-task class incremental setting, we use Hsu et al. [6]'s code to train the correspond models (the number for LwF is currently missing as it is not implemented in their code base; we plan to add the corresponding/missing entry in Table 3 for the final version of this paper).

Finally we also evaluate two meta-CL baselines: Online-aware Meta-Learning (OML; Javed and White [29]) and Generative Meta-Continual Learning (GeMCL; Banayeeanzade et al. [31]). OML is a MAML-based meta-learning approach. We note that as reported by Javed and White [29] in their public code repository; after some critical bug fix, the performance of their OML matches that of Beaulieu et al. [30] (which is a direct application of OML to another model architecture). Therefore, we focus on OML as our main MAML-based baseline. We take the out-of-the-box model (meta-trained for Omniglot, with a 1000-way output) made publicly available by Javed and White [29]. We evaluate the corresponding model in two ways. In the first, 'out-of-the-box' case, we take the meta-pre-trained model and only tune its meta-testing learning rate (which is done by Javed and White [29] even for meta-testing in Omniglot). We find that this setting does not perform very well; in the other case ('optimized # meta-testing iterations'), we additionally tune the number of meta-test training iterations. We've done a grid search of the meta-test learning rate in $3 * \{1e^{-2}, 1e^{-3}, 1e^{-4}, 1e^{-5}\}$ and the number of meta-test training steps in $\{1, 2, 5, 8, 10\}$ using a meta-validation set based on an MNIST validation set (5 K held-out images from the training set); we found the learning rate of $3e^{-4}$ and 8 steps to consistently perform the best in all our settings. We've also tried it 'with' and 'without'

Table 7: Impact of the number of in-context examples. Classification accuracies (%) on **Split-MNIST** in the 2-task and 5-task class-incremental learning (CIL) settings and the 5-task domain-incremental learning (DIL) setting. For ACL models, we use the same number of examples for meta-validation as for meta-training. According to Banayeeanzade et al. [31], GeMCL is meta-trained with the 5-shot setting but meta-validated in the 15-shot setting.

| Number of Examples | | DIL | | CIL 2-task | | CIL 5-task | |
| --- | --- | --- | --- | --- | --- | --- | --- |
| Meta-Train/Valid | Meta-Test | GeMCL | ACL | GeMCL | ACL | GeMCL | ACL |
| 5 | 5 | - | $84.1 \pm 1.2$ | - | $93.4 \pm 1.2$ | - | $74.6 \pm 2.3$ |
| | 15 | - | $83.8 \pm 2.8$ | - | $94.3 \pm 1.9$ | - | $65.5 \pm 4.0$ |
| 15 | 5 | $62.2 \pm 5.2$ | $83.9 \pm 1.0$ | $87.3 \pm 2.5$ | $93.6 \pm 1.7$ | $71.7 \pm 2.5$ | $76.7 \pm 3.6$ |
| | 15 | $\mathbf{63.8 \pm 3.8}$ | $\mathbf{84.5 \pm 1.6}$ | $\mathbf{91.2 \pm 2.8}$ | $\mathbf{96.0 \pm 1.0}$ | $\mathbf{79.0 \pm 2.1}$ | $\mathbf{84.3 \pm 1.2}$ |

the standard mean/std normalization of the MNIST dataset; better performance was achieved without such normalization (which is in fact consistent as they do not normalize the Omniglot dataset for their meta-training/testing). Their performance on the 5-task class-incremental setting is somewhat surprising/disappointing (since genenralization from Omniglot to MNIST is typically straightforward, at least, in common non-continual few-shot learning settings; see, e.g., Munkhdalai and Yu [51]). At the same time, to the best of our knowledge, OML-trained models have not been tested in such a condition in prior work; from what we observe, the publicly available out-of-the-box model might be overtuned for Omniglot/Mini-ImageNet or the frozen 'representation network' is not ideal for genenralization. We note that the sensitivity of these MAML-based methods [29, 30] w.r.t. meta-test hyper-parameters has been also noted by Banayeeanzade et al. [31]; these are characteristics of hand-crafted learning algorithms that we want to avoid with learned learning algorithms.

We use code and a pre-trained model (trained on Omniglot) made public by Banayeeanzade et al. [31] for the GeMCL baseline (see also Table 7); like our method, GeMCL also do not require any special tuning at test-time.

Our out-of-the-box ACL models (trained on Omniglot and Mini-ImageNet) do not require any tuning at meta-test time. Nevertheless, we've checked the effect of the number of meta-test training examples (5 vs. 15; 15 is the number used in meta-training); we found the consistent number, i.e., 15, to work better than 5. For the version that is meta-finetuned using the 5-task ACL objective (using only the Omniglot dataset), we use 5 or 15 examples for both meta-train and meta-test training (see an ablation study in Table 7). To obtain a sequence of 5 tasks, we simply sample 5 tasks from Omniglot (in principle, we should make sure that different tasks in the same sequence have no class overlap; in practice, our current implementation simply randomly draws 5 independent tasks from Omniglot).

### A.8 Details of the Split-CIFAR100 and 5-datasets experiment using ViT

As we described in Sec. 4, for the experiments on Split-CIFAR100 and 5-datasets, following Wang et al. [33, 34], we use ViT-B/16 pre-trained on ImageNet [76] which is available through `torchvision` [111]. In this experiments, we resize all images to 3x224x224 and feed them to the ViT. We remove the output layer of the ViT, and use its 768-dimensional feature from the penultimate layer as the image encoding. The self-referential component which is added to this encoder has the same architecture (2 layers, 16 heads) as the rest of the paper (see all hyper-parameters in Table 5) All ViT parameters are frozen during meta-training.

## B  Extra Experimental Results

### B.1  Ablation Studies on the Meta-validation Dataset

Here we conduct ablation studies on the choice of meta-validation sets to select model checkpoints. In general, when dealing with out-of-domain generalization, the choice of validation procedures to select final model checkpoints plays a crucial role in the evaluation of the corresponding method [112, 113]. The out-of-the-box models are chosen based on the average meta-validation performance on the validation set corresponding to the few-shot learning datasets used in meta-training: Omniglot and

Table 8: Meta-testing on sequences that are longer than those from meta-training. Classification accuracies (%) on 5-task **Split-FMNIST** and 5-task **Split-MNIST** in the **domain-incremental** settings. The model is the one finetuned with 5-task ACL loss using Omniglot as the meta-finetuning set and FMNIST as the meta-validation set (i.e., the numbers in the top part of the table are taken from Table 6). In the first column, "Split-FMNIST, Split-MNIST" indicates continual learning of 5 Split-FMNIST tasks followed by 5 tasks of Split-MNIST (and "Split-MNIST, Split-FMNIST" is the opposite order). Performance is measured at the end of the entire sequence.

| | | Meta-Test Test Tasks | |
| Meta-Test Training Task Sequence | # Tasks | Split-FMNIST | Split-MNIST |
| --- | --- | --- | --- |
| Split-FMNIST | 5 | $90.4 \pm 0.5$ | - |
| Split-MNIST | 5 | - | $81.6 \pm 1.3$ |
| Split-FMNIST, Split-MNIST | 10 | $79.3 \pm 2.7$ | $74.3 \pm 0.9$ |
| Split-MNIST, Split-FMNIST | 10 | $78.1 \pm 3.1$ | $78.5 \pm 1.7$ |

Table 9: Classification acuracies (%) on 5-task 2-way Split-Omniglot. Mean/std is computed over 10 meta-test runs.

| Method | Domain Incremental | Class Incremental |
| --- | --- | --- |
| GeMCL | $64.6 \pm 9.2$ | $97.4 \pm 2.7$ |
| ACL | $92.3 \pm 0.4$ | $96.8 \pm 0.8$ |

mini-ImageNet (or Omniglot, mini-ImageNet, and FC100 in the case of 3-task ACL), independently of any potential meta-test datasets. In contrast, in the meta-finetuning process of Table 3, we selected our model checkpoint by meta-validation on the MNIST validation dataset (we held out 5 K images from the training set). Here we evaluate ACL models meta-finetuned for the "5-task domain-incremental binary classification" on three Split-'X' tasks where 'X' is MNIST, FashionMNIST (FMNIST) or CIFAR-10 for various choices of meta-validation sets (in each case we hold out 5 K images from the corresponding training set). In addition, we also evaluate the effect of meta-finetuning datasets (Omniglot only v. Omniglot and mini-ImageNet). Table 6 shows the results (we use 15 meta-training and meta-testing examples except for the Omniglot-finedtuned/MNIST-validated model from Table 3 which happens to be configured with 5 examples; this will be fixed in the final version). Effectively, meta-validation on the matching validation set is useful. Also, meta-finetuning only on Omniglot is beneficial for the performance on MNIST when meta-validated on MNIST or FMNIST. However, importantly, we emphasize that our ultimate goal is not to obtain a model that is specifically tuned for certain datasets; we aim at building models that generally work well across a wide range of tasks (ideally on any tasks); in fact, several existing works in the few-shot learning literature evaluate their methods in such settings (see, e.g., Requeima et al. [114], Bronskill et al. [78], Triantafillou et al. [115]). This also goes hand-in-hand with scaling up ACL (our current model is tiny; see hyper-parameters in Table 5; the vision component is also a shallow 'Conv-4' net) and various other considerations on self-improving continual learners (see, e.g., Schmidhuber [116]), such as automated curriculum learning [117].

## B.2 Performance on Split-Omniglot

Here we report the performance of the models used in the Split-MNIST experiment (Sec. 4.3) on "in-domain" 5-task 2-way Split-Omniglot. Table 9 shows the result. Performance is very similar between our ACL and the baseline GeMCL on this task in the class incremental setting, unlike on Split-MNIST (Table 3) where we observe a larger performance gap between these same models. Here we also include the "domain incremental" setting for the sake of completeness but note that GeMCL is not originally trained for this setting.

Table 10: 5-way classification accuracies using 15 examples for each class for each task in the context. 2-task models are meta-trained on Omniglot and Mini-ImageNet, while 3-task models are in addition meta-trained on FC100. 'A, B' in 'Context/Train' column indicates that models sequentially observe meta-test training examples of Task A then B; evaluation is only done at the end of the sequence. "no ACL" is the baseline 2-task models trained without the ACL loss.

| Meta-Testing Tasks | | Number of Meta-Training Tasks | | |
|---|---|---|---|---|
| Context/Train | Test | 2 (no ACL) | 2 | 3 |
| A: MNIST-04 | A | $71.1 \pm 4.0$ | $75.4 \pm 3.0$ | $89.7 \pm 1.6$ |
| B: CIFAR10-04 | B | $51.5 \pm 1.4$ | $51.6 \pm 1.3$ | $55.3 \pm 0.9$ |
| C: MNIST-59 | C | $65.9 \pm 2.4$ | $63.0 \pm 3.3$ | $76.1 \pm 2.0$ |
| D: FMNIST-04 | D | $52.8 \pm 3.4$ | $54.8 \pm 1.3$ | $59.2 \pm 4.0$ |
| | Average | 60.3 | 61.2 | 70.1 |
| A, B | A | $43.7 \pm 2.3$ | $81.5 \pm 2.7$ | $88.0 \pm 2.2$ |
| | B | $49.4 \pm 2.4$ | $50.8 \pm 1.3$ | $52.9 \pm 1.2$ |
| | Average | 46.6 | 66.1 | 70.5 |
| A, B, C | A | $26.5 \pm 3.2$ | $64.5 \pm 6.0$ | $82.2 \pm 1.7$ |
| | B | $32.3 \pm 1.7$ | $50.8 \pm 1.2$ | $50.3 \pm 2.0$ |
| | C | $56.5 \pm 8.1$ | $33.7 \pm 2.2$ | $44.3 \pm 3.0$ |
| | Average | 38.4 | 49.7 | 58.9 |
| A, B, C, D | A | $24.6 \pm 2.7$ | $64.3 \pm 4.8$ | $78.9 \pm 2.3$ |
| | B | $20.6 \pm 2.3$ | $47.5 \pm 1.0$ | $49.2 \pm 1.3$ |
| | C | $38.5 \pm 4.4$ | $32.7 \pm 1.9$ | $45.4 \pm 3.9$ |
| | D | $36.1 \pm 2.5$ | $31.2 \pm 4.9$ | $30.1 \pm 5.8$ |
| | Average | 30.0 | 43.9 | 50.9 |

### B.3 Effect of Number of In-Context Examples

Table 7 shows an ablation study on the number of examples used for meta-training and meta-testing on the Split-MNIST task. We observe that for an ACL model trained only with 5 examples during meta-training, more examples (15 examples) provided during meta-testing is not beneficial. In fact, they even largely hurt in certain cases (see the last column); this is one form of "length generalization" problem. When the number of meta-training examples is consistent with the one used during meta-testing, the 15-example case consistently outperforms the 5-example one.

### B.4 Effect of Number of Tasks in the ACL Loss

Table 10 provides the complete results discussed in Sec. 4.3 under "Evaluation on diverse task domains".

### B.5 Further Discussion on Limitations

Here we provide further discussion and experimental results on the limitations of our approach as a learned algorithm.

**Domain generalization.** As a data-driven learned algorithm, the domain generalization capability is a typical limitation as it depends on the meta-trained data. Certain results we presented above are representative of this limitation. In particular, in Table 6, the model meta-trained/finetuned on Omniglot using Split-MNIST as meta-validation set do not perform well on Split-CIFAR10. While meta-training and meta-validating on a larger/diverse set of datasets may be an immediate remedy to obtain more robust ACL models, we note that since ACL is also a "continual meta-learning" algorithm (Sec. 5), an ideal ACL model should also continually incorporate and learn from more data during potentially lifelong meta-testing; we leave such an investigation for future work.

**Length generalization.** We already qualitatively observed the limited length generalization capability in Table 10 (meta-trained with up to 3 tasks and meta-tested with up to 4 tasks). Here we provide one more experiment evaluating ACL models meta-trained for 5 tasks on a concatenation of two 5-task Split-MNIST and Split-FMNIST tasks (resulting in 10 tasks). Table 8 shows the results. Again,

while the model does not completely break, increasing the number of tasks to 10 rapidly degrades the performance compared to the 5-task setting the model is meta-trained for. Similarly, its performance on the Split-Omniglot domain incremental setting (Sec. B.2) degrades with increased numbers of tasks: accuracies for 5, 10 and 20 tasks are $92.3\% \pm 0.4$, $82.0\% \pm 0.4$ and $67.6\% \pm 1.1$ respectively. As noted in Sec. 5, this is a general limitation of sequence processing neural networks, and there is a potential remedy for this limitation (meta-training on more tasks and "context carry-over") which we leave for future work.

## B.6 A Comment on Meta-Generalization

We also note that in general, "unseen" datasets do not necessarily imply that they are harder tasks than "in-domain" test sets; when meta-trained on Omniglot and mini-ImageNet, meta-generalization on "unseen" MNIST is easier (the accuracy is higher) than on the "in-domain" test set of mini-ImageNet with heldout/unseen classes (compare Tables 1 and 2).

