# OpenReview forum: "Metalearning to Continually Learn In Context"
_NeurIPS.cc/2024/Conference — Submitted to NeurIPS 2024_

### Official Review · Reviewer_vTAU · 2024-07-07

**Soundness:** 3
**Presentation:** 3
**Contribution:** 3
**Rating:** 5
**Confidence:** 4

**Summary:**

The paper focuses on Automated Continual Learning which is different than handcrafted continual learning. It uses self referential neural networks to meta learn their own in-context continual learning algorithm. First, the paper shows the emergence of in-context catastrophic forgetting. Second, the paper analyze the performance of proposed method (ACL) and finally the paper discuss the limitation of the proposed method.

**Strengths:**

- The paper is clearly written and easy to follow
- The paper introduces original idea of Automated Continual Learning
- The paper identifies "in-context" catastrophic forgetting

**Weaknesses:**

- The paper claims to do in-context continual learning but the concept of in-context learning is not clearly explained.
- The paper mainly focus on two task and five task settings but it would be more helpful to see the more different settings such as three task or four task
- How is the size of SRWM affects the maximum sequence length that can be train?

**Questions:**

- Why only consider the two-task setting?
- Why ACL was not compared with replay buffer based methods?
- What is the architecture of SRWM?

**Limitations:**

Authors have addresses the limitation of the work.

---

> ### Author Rebuttal · Authors · 2024-08-06
>
> We would first like to thank the reviewer for their valuable time reviewing our work and for many positive comments. Thank you very much.
>
> > The paper claims to do in-context continual learning but the concept of in-context learning is not clearly explained.
>
> We actually describe and highlight the concept of in-context learning in Sec 2.2 and Figure 1. But we agree with the reviewer that this is currently not clear as we also use the somewhat older terminology of “Meta-learning via Sequence Learning” in the title of Sec 2.2 (we explain much later in Line 337 “This was rebranded as in-context learning” in Sec. 5). We believe mentioning “in-context learning” in Sec 2.2 upfront should make this context clearer. We will fix this in the final version. Thank you very much for pointing this out.
>
> > The paper mainly focus on two task and five task settings but it would be more helpful to see the more different settings such as three task or four task
>
> We actually provide many more experiments in the appendix for the interested readers. Table 10 provides results for 2, 3, and 4 tasks (exactly as suggested by the reviewer) and Table 8 explores generalization up to 10 tasks (by concatenating Split-MNIST and Split-FashionMNIST).
>
> > How is the size of SRWM affects the maximum sequence length that can be train?
>
> This is a hard question for which we do not have a straightforward answer because this depends a lot on the nature of the datasets. That said, one useful theoretical result to keep in mind is that, given the dimension of weight matrix D-by-D, the maximum number of key-value associations one can retrieve in a noiseless fashion is D (corresponding to the maximum number of orthogonal keys one can have in the D dimensional space). In practice, the notion of capacity is more complex as the model has multiple layers, and depending on the nature of the data, certain keys may be shared by different inputs without loss of performance.
>
> > Why only consider the two-task setting?
>
> As we mentioned above, we consider many more settings than the two-task setting: five-task setting (Split-MNIST) in Table 3, two/three/four tasks in Table 10, and up to ten tasks in Table 8.
>
> Perhaps what the reviewer really meant is: why do we first focus on the 2-task setting? The reason is clear: Our motivation for using two tasks in Sec. 4.1 and 4.2 is to introduce the core problem of in-context catastrophic forgetting and to demonstrate how our ACL overcomes this problem in a *minimum* and *comprehensible* setting (and that's the 2-task setting).
>
> > Why ACL was not compared with replay buffer based methods?
>
> We focus on the replay-free setting as we believe this is the most interesting setting in continual learning currently (allowing for eliminating replay buffers which requires extra engineering design regarding what to store/remove, and to manage extra memory storage of raw data). Similarly, the recent "learning-to-prompt" papers we cite in our work also focus on the replay-free setting. They additionally raise the privacy issue of the replay buffer which stores raw data to motivate the replay-free setting.
>
> Please also note that our ACL and the use of replay buffer are orthogonal: they could also be combined together.
>
> > What is the architecture of SRWM?
>
> The architectural details of SRWM can be found in Table 5 in the appendix.
>
> We believe our response thoroughly clarifies all the reviewer's remaining concerns. Please also refer to our general response highlighting our contributions that we believe are highly relevant considering the diverse interests represented in the NeurIPS community.
> We really believe our contributions outweigh the limitations, and the current overall rating does not fully reflect the contributions of this paper.
> If you think that our rebuttal has further improved the reviewer's perception and rating of our work, we would appreciate it a lot if the reviewer could consider increasing the score. Thank you very much.

---

> ### Comment · Reviewer_vTAU · 2024-08-12
> **Update**
>
> Thank you for the detailed explanation. I have missed major points raised by other reviewers and feel like I should change my original review. I believe that the paper presents very original idea and need just final finishing in terms of clarity.

---

> > ### Author Response · Authors · 2024-08-12
> > **Explanation requested**
> >
> > Dear Reviewer vTAU,
> >
> > Thank you for your response.
> >
> > However, given that you changed your score from 6 to 4, we would like to know more elaborated explanations/justifications, especially considering your high/influential confidence score of 4.
> >
> > You wrote:
> >
> > > I have missed major points raised by other reviewers
> >
> > What "major points" are you referring to? As we explained in our rebuttal, we tried to resolve many concerns raised by the reviewers, including corrections of certain factual misunderstandings. They have not responded yet, but we hope they will.
> >
> > We would like to express our concern that for now, our rebuttal has not been considered. This is very unfair.
> >
> > If you found that our response did not convincingly address these points, please explain the reasons.

---

> > > ### Author Response · Authors · 2024-08-13
> > >
> > > We really do not intend to bother the reviewer further, but we wanted to let you know that Reviewer d5XY has increased the score after considering our rebuttal... If you think their original review had influenced your score change, please do not hesitate to take a look at their response.

---

> > > > ### Author Response · Authors · 2024-08-14
> > > > **Thank you**
> > > >
> > > > We acknowledge the increased score and appreciate your consideration. Thank you very much.

---

### Official Review · Reviewer_x6Jn · 2024-07-12

**Soundness:** 4
**Presentation:** 4
**Contribution:** 3
**Rating:** 5
**Confidence:** 4

**Summary:**

The paper describes a method for in-context continual learning (CL) by using a type of meta-learning neural architecture based on ‘self-referential weight matrices’ (SRWM). Proposed in prior work, these models learn to modify weight matrices iteratively as they process more and more inputs. In this work, they are given few-shot examples from different tasks and iteratively update the weight matrices as the examples are processed. This update process is referred to as “in-context” learning in this work. The key innovation is to define the loss function of SRWM training to optimise for both forward (improving performance of subsequent CL tasks) and backward (improving performance of previous CL tasks) transfer while achieving good performance on the current task. Experiments are conducted on commonimage classification meta-learning benchmarks such as Split-MNIST and Mini-ImageNet. Results show the proposed method prevents catastrophic forgetting (without using replay), outperforming existing meta-learning baselines on the evaluated benchmarks.

**Strengths:**

Studies the problem of in-context catastrophic forgetting via a two-task toy setting and reveals the issue when training with no backward transfer loss term. This is shown to be mitigated by including the backward transfer loss term.

Proposes an in-context CL method using models based on SRWM and a novel loss to mitigate catastrophic forgetting as more tasks are learned. The method does not use a replay buffer.

Studies and covers standard image classification meta-learning tasks such as Split-MNIST, FashionMNIST, and CIFAR-10. On Split-MNIST, shows improvements over existing CL and meta-baselines in both domain and class incremental evaluation settings. The improvements, when additional 5-task fine-tuning is used, is significantly above baselines.

The paper is clearly written, with thorough literature review.

**Weaknesses:**

One weakness of the proposed method is that the number of loss function terms increases with the number of CL tasks, as pointed out by the authors in Appendix A.5. This prevents this method from being scaled to more practically relevant settings where a large number (much more than 2 or 3 that this paper has mostly focused the experiments on) of tasks are considered in a CL setting. Method of reducing the loss terms would strengthen the paper.

Another weakness, which is also noted by the authors in Table 4 and Section 4.3, is that the performance of the proposed model and method is poor compared with those based on pre-trained transformer models, even on an easier evaluation task. The authors in Section 5 also discuss a potential connection between LLM transformer training as an implicit version of the proposed model and method. Given these existing strong and more widely adopted methods, it is unclear how much value the proposed method adds. SRWMs are not widely used and LLMs training can scale to a massive number of tasks with a single loss [1] (albeit not CL). A more detailed explanation of the application of the findings of this paper beyond those interested in SRWMs would be helpful.

Another weakness of this paper is its focus on image classification meta-learning tasks only. It is helpful to know the generality of this method, for example on language modelling tasks or multimodal tasks. An experiment demonstrating the method in CL language tasks would be helpful.

[1] Finetuned language models are zero-shot learners. Wei et al. ICLR 2022.

**Questions:**

None

**Limitations:**

Limitations have been adequately addressed.

---

> ### Author Rebuttal · Authors · 2024-08-06
>
> We thank the reviewer for their valuable time reviewing our work and for many positive comments.
>
> We also acknowledge the reviewer’s thorough reading through the details of our work. Thank you very much.
>
> > One weakness of the proposed method is that the number of loss function terms increases with the number of CL tasks, as pointed out by the authors in Appendix A.5. … Method of reducing the loss terms would strengthen the paper.
>
> Yes, the discussion on this limitation is provided in A.5./L768. Given that the most crucial loss terms are those at the last time step of the sequence, one possibility to reduce the number of terms is to sample a subset of the intermediate terms. However, effectively validating such methods would require actual experiments involving many more tasks, which we leave for future work.
>
> > Another weakness, which is also noted by the authors in Table 4 and Section 4.3, is that the performance of the proposed model and method is poor compared with those based on pre-trained transformer models
>
> Yes, clearly we will need meta-training on larger and more diverse datasets for this method to demonstrate its full potential. In fact, as a purely data-driven method, more and more improvements are potentially expected by scaling this up with more compute and datasets (Sutton’s bitter lesson).
>
> > it is unclear how much value the proposed method adds. SRWMs are not widely used and LLMs training can scale to a massive number of tasks with a single loss [1] (albeit not CL). A more detailed explanation of the application of the findings of this paper beyond those interested in SRWMs would be helpful.
>
> Thank you very much for asking this question. Here we would like to bring up one important and (hopefully) convincing aspect. It is true that SRWM itself is not widely used at the moment. That said, there is an emerging trend in sequence processing using fast weights (a family of linear Transformers). As we also emphasized in the general response, two very recent works [2] and [3] show very promising results on general language modeling using such architectures.
>
> It should be noted that [2] makes use of DeltaNet [1], whose direct successor is SRWM (DeltaNet augmented by self-reference), and [4] shows that SRWM is indeed more powerful than DeltaNet (using formal languages) while not requiring hard-to-parallelize, true recurrence. Besides, [3] also mentions “multi-level meta-learning” in the outlook, which was essentially the original motivation of SRWM.
>
> Given these evidences of active research on this family of models, in our view, it is not unlikely that there will soon be other works scaling up SRWM or similar models on other tasks. Here we motivated SRWM as a natural architecture for continual learning but its applicability is broader in principle (maybe similar to how attention/Transformer was first explored in machine translation as a natural architecture). From this perspective, we believe we have one extra contribution in this work that is broader than the specific scope of this work. This is a successful example of sequence processing using SRWM which may also be of interest to those interested in general sequence processing using fast weights or linear Transformers. We believe this point precisely addresses the reviewer's concern regarding the impact of this work beyond our specific scope.
>
> > Another weakness of this paper is its focus on image classification meta-learning tasks only.
>
> This is another very valid point. As a first paper on this method, we focused on image classification which has classic continual learning benchmarks. Extending this to other modalities is an exciting future research (we also mention this in L311/Sec 5).
>
> We hope this response and the general response further clarifies our contributions (including the relevance of SRWM in a broader scope of sequence processing with fast weights).  If you think our response has enhanced your perception of this work, and if you think the paper should be accepted, we would appreciate it a lot if the reviewer could consider increasing the score. Thank you very much.
>
> [1] Schlag et al. ICML 2021. Linear transformers are secretly fast weight programmers. https://arxiv.org/abs/2102.11174
>
> [2] Yang et al. arXiv June 2024. Parallelizing Linear Transformers with the Delta Rule over Sequence Length.  https://arxiv.org/abs/2406.06484
>
> [3] Sun et al. arXiv July 2024. Learning to (Learn at Test Time): RNNs with Expressive Hidden States. https://arxiv.org/abs/2407.04620
>
> [4] Irie et al. EMNLP 2023. Practical Computational Power of Linear Transformers and Their Recurrent and Self-Referential Extensions. https://arxiv.org/abs/2310.16076

---

> > ### Comment · Reviewer_x6Jn · 2024-08-13
> > **Comment**
> >
> > I thank the authors for the detailed rebuttal. It appears that a majority of the concerns mentioned in the review has been considered by the authors but as future work (for example whether subsampling the loss terms for more tasks can work, or expanding beyond the image domain). The contributions of this paper would be much stronger for these results to be included in this paper. Given the existing results, the generality of this method remains quite unclear. I maintain my original recommendation.

---

> > > ### Author Response · Authors · 2024-08-13
> > > **Thank you for your response**
> > >
> > > Thank you very much for your response.
> > >
> > > We agree that the paper will be much stronger if we could solve all these problems.
> > >
> > > However, it is unreasonable to assume that one can solve all the problems in this single paper.
> > >
> > > It is not as if our paper lacks content; **we have a full-length paper with plenty of novel results.**
> > >
> > > We are deeply disappointed that the reviewers seem to consider our results/contributions to be trivial, and only evaluate our methods through their limitations (when many of them can be solved directly by better-resourced teams).

---

### Official Review · Reviewer_HvJH · 2024-07-13

**Soundness:** 3
**Presentation:** 2
**Contribution:** 3
**Rating:** 5
**Confidence:** 3

**Summary:**

The paper studies the problem of catastrophic forgetting (CF) by formulating continual learning (CL) as learning from a sequence of demonstrations of tasks. The paper proposes a meta-learning objective function that includes backward transfer terms. These terms compute the error of the predictor on previous tasks after receiving demonstrations of the current task.

**Strengths:**

- The approach of formulating (continual learning) CL as learning from a sequence of demonstrations of tasks is interesting.
- The experiment shows positive results when compared to non-meta-learning approaches

**Weaknesses:**

- The paper is difficult to follow. Many definitions and the algorithm are not very well explained.
    - The motivation of formulating (continual learning) CL as meta-learning is not well presented.
    - Some details of the architecture are mentioned in the background section only (e.g. replacing self-attention with SRWN and the multi-head version.)
    - The details of the training and inference process are not well presented.
- The training process can be very costly and poorly scaled with the number of tasks and the number of examples per task. In each step over a sequence of demonstrations, the method needs to compute and store a new weight matrix in order to perform back-propagation. It might require more memory during training and at inference.
- Even being a meta-learning approach, the model still needs fine-tuning when given a new task to adapt to a new number of tasks.

**Questions:**

- Can the authors explain more on the following claim “The stability-plasticity dilemma are automatically discovered and handled by the gradient-based program search process.“ (line 52)?
- What are the advantages of this method compared to previous approaches?
- How do the number of examples per task and order of tasks during the training affect the performance at inference time?
- How does the method scale with the number of tasks in terms of performance and computation?
- It’s unclear how to calculate the loss function in a batch fashion since each training point requires a different sequence of inputs (depending on the position of the task in the sequence) and loss components.

**Limitations:**

There are no negative social impacts. My suggestions have been listed in the previous sections.

---

> ### Author Rebuttal · Authors · 2024-08-06
>
> We thank the reviewer for their valuable time spent on reviewing our work.
>
> We believe we have good responses to resolve all the main concerns.
>
> **== Factual clarifications ==**
>
> Before providing our clarifications to the reviewer’s concerns, we would first like to resolve some factual misunderstandings.
>
> >  In each step over a sequence of demonstrations, the method needs to compute and store a new weight matrix in order to perform back-propagation. It might require more memory during training and at inference.
>
> This is not correct. There is no need for such storage (there is a memory-efficient algorithm that is standard in any practical linear Transformer implementations). For a detailed explanation, please kindly refer to our response to Reviewer d5XY (marked "**[also relevant to Reviewer HvJH]**"). We apologize for this inconvenience due to the length limit.
>
> > The experiment shows positive results when compared to non-meta-learning approaches
>
> Thank you for mentioning this as a strength. We would like to add that positive results are also shown compared to the *existing meta-learning methods* (Table 3).
> Unlike prior meta-learning methods that only learn “representations” for CL, our method successfully learns an entire “learning algorithm” (the original definition of learning-to-learn) and outperforms prior methods. We believe this achievement in meta-learning is largely overlooked in the current review.
>
> **== Clarifications to weaknesses/questions ==**
>
> > The motivation of formulating (continual learning) CL as meta-learning is not well presented.
>
> The motivation is described in the introduction (Line 30-50). Generally speaking, machine learning is useful when it is hard for humans to design a hand-crafted solution. Here we claim that hand-crafting CL algorithms has been unsuccessful in the past. Therefore, we propose to use machine learning to automate the process of designing CL algorithms. This corresponds to “learning of learning algorithms”, or meta-learning.
>
> > Some details of the architecture are mentioned in the background section only (e.g. replacing self-attention with SRWN and the multi-head version.)
>
> We’d like to clarify that SRWM is indeed a background work (including its role as a replacement to self-attention and the multi-head version). These design details are entirely based on the previous work by Irie et al. ICML 2022 "A Modern Self-Referential Weight Matrix That Learns to Modify Itself".
>
> > The details of the training and inference process are not well presented.
>
> The corresponding processes are described in Sec 3. and visually highlighted in Figure 1.
> If the reviewer still thinks these are “not well presented”, we would appreciate it a lot if they could tell more concretely what they find confusing. (A dedicated section in Appendix A.1 “Continual and Meta-learning Terminologies” also helps readers with continual and meta-learning jargon).
>
> > Even being a meta-learning approach, the model still needs fine-tuning when given a new task to adapt to a new number of tasks
>
> This is a valid point and it remains an open research question e.g., to find how to deal with an unseen maximum number of classes in the class-incremental setting. Here we would like to strongly emphasize that meta-learning is an open research topic. **We can not solve all these problems in a single paper.** A significant achievement here is that we obtain a successful meta-learned CL algorithm for a fixed maximum number of classes. Extending our work to relax these constraints is an exciting future research direction in meta-learning.
>
> > Can the authors explain more on the following claim “The stability-plasticity dilemma are automatically discovered and handled by the gradient-based program search process.“ (line 52)?
>
> Yes, our pleasure! When we manually design a CL algorithm (say a regularization method with a hyper-parameter weighting the auxiliary term), we face the problem of deciding how much we allow model parameters to change (plasticity) or not (stability) to balance preservation of current knowledge against learning of new one. In our method, we do not have to deal with this decision ourselves. This process is automated by gradient descent that directly optimizes weight modification rules (Eq. 3) for the ultimate model performance.
>
> > What are the advantages of this method compared to previous approaches?
>
> > How does the method scale with the number of tasks in terms of performance and computation?
>
> The current advantage is that we can achieve a CL algorithm that performs well on classic benchmarks (unlike prior methods that fail). Another potential advantage is that, as a purely data-driven method, more and more improvements are expected by scaling this up with more compute and datasets (Sutton’s bitter lesson). Telling exactly how they scale would require a proper scaling law study, itself requiring more compute.
>
> > How do the number of examples per task and order of tasks during the training affect the performance at inference time?
>
> An ablation on the number of examples can be found in Table 7 in the appendix (5 vs. 15 examples). Regarding the order, we conducted experiments in the 2-task (A and B) setting: Tables 1 and 2 present results for both A-then-B and B-then-A orders.
>
> > It’s unclear how to calculate the loss function in a batch fashion
>
> There is no such issue: each sequence in the batch is constructed such that it contains the same number of examples and the task boundaries at the same time steps.
>
> We believe our response thoroughly addresses all the concerns raised by the reviewer. Please also refer to our general response highlighting our contributions that we believe are broadly relevant to the NeurIPS community, despite all our limitations. In light of these clarifications, we believe the current rating does not fully reflect the contributions of this paper. If the reviewer finds our response useful/convincing, please consider increasing the score. Thank you very much.

---

> > ### Author Response · Authors · 2024-08-14
> > **Author-reviewer discussion ending imminently**
> >
> > While we have been respecting the rule of not sending individual reminders to the reviewers ourselves, please allow us to post this single/final message as the end of author-reviewer discussion period is imminent.
> >
> > We would like to thank the reviewer one more time for their valuable time spent on reviewing our work.
> >
> > As all the other reviewers have already responded, we would appreciate it a lot if the reviewer could communicate their perception and rating of this work that take into account our rebuttal.
> >
> > Thank you very much.

---

> > ### Comment · Reviewer_HvJH · 2024-08-14
> > **Response to Authors' Rebuttal**
> >
> > Thank you for your detailed rebuttal. It really helps clarify my concerns. I agree that the formulation of CL as an ICL problem is interesting and the fact that the trained model can generalize is even more impressive. I am raising the score to 5 and am willing to increase it if more information/discussion is provided.
> >
> > 1. Can you discuss the importance/advantage/limitation of the self-referential weight matrix in facilitating in-context learning? Self-referential is a powerful idea but it is not very clear why you choose that algorithm for CL.
> > 2. At first glance, the update rule for $W_t$ is quite simple, is it general enough to learn all kinds of algorithms? This question might be out of the scope of this paper but I just want to have a conversation.
> >
> > Also, I still maintain that the presentation of this paper could be improved.

---

### Official Review · Reviewer_d5XY · 2024-07-15

**Soundness:** 3
**Presentation:** 3
**Contribution:** 2
**Rating:** 5
**Confidence:** 4

**Summary:**

The paper proposes a novel technique to automatically discover in-context continual learning dynamics for image classification task sequences through meta-learning. In order to achieve this purpose, the approach relies on 2 main novelties:
* Using self referential weight matrices on top of an image encoder - SRWM, as self-modifying that adapts itself to the stream of inputs, is an natural model for continual learning.
* Encoding continual learning desiderata in the meta-objective, i.e. backward and forward transfer.

The authors first apply the approach in a classic two-task setting (Split-MNIST) that allows them to showcase and analyse the emergence of in-context catastrophic forgetting phenomena, and to show that using their ACL loss can help reduce it. They further evaluate their method and compare them to replay-free baselines from the CL and meta-CL literature, showing an advantage of their approach in scenarios with up to 3 tasks.

The authors further test the limits of their approach by comparing it to more recent learning to prompt techniques for continual learning, leveraging the power of pretrained large models. This scenario show a limitation of the technique in more complex scenarios with more tasks, more diverse and complex data.

**Strengths:**

* The paper takes an interesting perspective on continual learning, leveraging the interesting properties of SRWM and the capability of meta-learning to encode the desired behavior in the meta-learning objective. The combination of these two contributions is novel to the best of my knowledge, and lead to interesting insights.

* The approach leads to interesting performance in relatively simple scenarios, outperforming some of the existing continual learning techniques.

* I also particularly appreciated the authors discussion of the method limitations. Both the experiments with learning to prompts and the discussion provide very valuable insights that can help building on the work in the future.

**Weaknesses:**

* In my opinion, the main limitation of the approach is its practicality. From the experiments reported in Table 4, it seems that the approach requires to met-train on a sequence of similar length and/or complexity to provide its potential. This is not possible to know in advance in practice. Moreover, one limitation that the authors have not mentioned is that the meta-objective seems to require keeping in memory a number of copies of the model that is equal to the number of tasks. This can quickly become cumbersome for real applications that can require more complex models and very long sequences of tasks.

* While the authors focus on classic benchmarks for continual and meta-learning, these benchmarks are artificial, relatively simple and lack of diversity. Different works highlight the limits of these benchmarks, I invite the authors to look at "Meta-Album: Multi-domain Meta-Dataset for Few-Shot Image Classification" Ullah et al. 2023, and "NEVIS'22: A Stream of 100 Tasks Sampled from 30 Years of Computer Vision Research" Bornschein et al. 2023 for examples of more realistic benchmarks.

* It would be interesting to add a discussion of the cost of the approach (computation, memory, ...). Even is it gives a substantial boost in many cases, it would be interesting for practitioners to compare what they gain to what they pay.

**Questions:**

* The approach is focused on the task aware scenario, and rooted in a notions of tasks. In many practical scenarios, the distribution shift occurs in a softer way, with no clear notion of task boundaries. Can the authors comment on the possibility of extend their approach to the task-agnostic scenario?

**Limitations:**

The authors provide a detailed discussion of the work limitations, both in the experiments and the discussion sections. Some other limitations are highlighted in the Weaknesses paragraph above.

---

> ### Author Rebuttal · Authors · 2024-08-06
>
> We thank the reviewer for their valuable time reviewing our work and for many positive comments. Thank you very much.
>
> **== Factual error corrections ==**
>
> Before providing our clarifications to the reviewer’s concerns, we would first like to resolve some factual errors in the review.
>
> > in a classic two-task setting (Split-MNIST)
>
> >  showing an advantage of their approach in scenarios with up to 3 tasks.
>
> These statements are not correct. The main results on Split-MNIST (Table 3) correspond to a *5-task* setting.
> (We also report the generalization performance on a *10-task* setting by using Split-MINST and Split-FashionMNIST in Table 8 in the appendix).
>
> > Moreover, one limitation that the authors have not mentioned is that the meta-objective seems to require keeping in memory a number of copies of the model that is equal to the number of tasks.  **[also relevant to Reviewer HvJH]**
>
> This is not correct. We do not need to keep the intermediate copies of the model. The reviewer is right to point out that a *naive* implementation would require such copy storage. The situation is actually even worse for a naive implementation: it would have to store ALL intermediate copies of the model’s fast weight states (at every step) for backpropagation through time; this would result in impractical memory requirements for ANY experiments we conducted.
>
> Instead, the practical code implements a custom memory-efficient algorithm that only stores one copy of the model fast weights in the entire forward pass, and during backpropagation through time, we de-construct the model by applying the reverse fast weight update (Eq.3 in the backward direction) to obtain the model state we need at the corresponding step. This only requires storing the key/value/query/learning-rate activations for all steps in memory (which are much cheaper) and a single copy of the model.
>
> We did not describe this implementation trick as this is standard in ANY practical linear Transformer implementations (in case the reviewer is interested, please find example public code below). We will add this discussion in the final version. Thank you for pointing this out.
>
> Public code references:
> - Linear Transformer: https://github.com/idiap/fast-transformers/blob/master/fast_transformers/causal_product/causal_product_cuda.cu
>
> - SRWM: https://github.com/IDSIA/modern-srwm/blob/main/supervised_learning/self_ref_v0/self_ref_v0.cu
>
> **== Clarifications to weaknesses/questions ==**
>
> We believe we have convincing answers to all the main concerns raised by the reviewers.
>
> > the main limitation of the approach is its practicality.
>
> We agree with the reviewer that our method requires scaling multiple hyper-parameters to be useful in more realistic settings. That said, our main bottleneck is the compute. In the limitation section, we do propose approaches to deal with the algorithmic limitations (e.g., to handle longer lengths, we’ll have to introduce context-carry over as used in language modeling with linear Transformers; see Line 302). There is no reason that better-resourced teams, which can afford to train today's large language models, cannot scale this up to much larger and more diverse datasets.
>
> > I invite the authors to look at …  examples of more realistic benchmarks.
>
> We thank the reviewer for sharing these references. We’d like to emphasize that it’s not that we didn’t know about these more realistic tasks, but rather that we do not have compute to conduct larger scale experiments. Please note that given that our method is novel, we had to allocate a lot of compute for ablation experiments (we have many more experiments in the appendix).
>
> As we also emphasized in our general response, we strongly believe that this work has significant contributions by providing several evidences for the promise of this method, even without a large scale continual learning experiments. These contributions are largely overlooked in the current review.
>
> > It would be interesting to add a discussion of the cost of the approach (computation, memory, ...). Even is it gives a substantial boost in many cases, it would be interesting for practitioners to compare what they gain to what they pay.
>
> Assuming that model hyper-parameters are already available, as we mention in A.3., less than 1 day of compute using a single V100 GPU is enough to produce a single-run experiment reported in this paper (e.g., the Split-MNIST result). We also would like to emphasize that our method does not just provide a “substantial boost” but rather represents a clear-cut switch from a complete failure (hand-crafted approach or some other meta-learning method) to reasonable continual learning in many cases.
>
> > The approach is focused on the task aware scenario, and rooted in a notions of tasks. In many practical scenarios, the distribution shift occurs in a softer way, with no clear notion of task boundaries. Can the authors comment on the possibility of extend their approach to the task-agnostic scenario?
>
> Thank you for pointing this out. The task awareness is only required for meta-training. At test time, the model is not aware of anything related to the task identities or task boundaries; they are evaluated in a completely task-agnostic manner. For training, it seems reasonable to assume that we do have access to training examples where we know task identities. Therefore, extending this to the task-agnostic setting is rather straightforward.
>
> We believe our response thoroughly addresses all the concerns raised by the reviewer, and directly resolves many of them. Please also refer to our general response highlighting our contributions that we believe are highly relevant considering the diverse interests represented in the NeurIPS community. In light of these clarifications, we believe the current rating does not fully reflect the contributions of this paper. If the reviewer finds our response useful/convincing, please consider increasing the score. Thank you very much.

---

> > ### Comment · Reviewer_d5XY · 2024-08-13
> > **Answer to rebuttal**
> >
> > I thank the authors for their clarifications:
> >
> > * "up to 3 tasks" in my original review is a typo - I meant 5 tasks, and I thank the authors for pointing out the additional results in the appendix. This however doesn't alleviate my main concern here: These validation benchmarks are still of limited length and diversity.
> >
> > * I thank the authors for the clarifications and pointers regarding the memory constraint. This answers my concern.
> >
> > In general, I find that the authors answered most of my questions. Despite the limitation in the evaluation benchmarks mentioned above, I am raising my score.

---

> > > ### Author Response · Authors · 2024-08-13
> > >
> > > Thank you very much for your reply and the increased score.
> > >
> > > We genuinely thank the reviewer for the effort put into checking our rebuttal.

---

### Author Rebuttal · Authors · 2024-08-06

**== General Response to all the Reviewers ==**

We would first like to sincerely thank all the reviewers for their valuable time reviewing our work.

We would like to emphasize that this work has *two facets*:
On the one hand, we explore a novel perspective/approach to *continual learning* (CL).
At the same time, this is also a *meta-learning* research paper presenting a significant advancement in this domain (learning of learning algorithms). This is a highly relevant topic to the NeurIPS community representing diverse interests, aiming an audience beyond those solely interested in state-of-the-art methods in CL.

We believe our contributions in meta-learning are largely overlooked in the current reviews.

In fact, we explicitly mention and acknowledge all the current limitations of our method in light of state-of-the-art continual learning methods/tasks. We did this honest exhibition not just to facilitate the reviewer’s job to identify these weaknesses; we did this because we strongly believe that, *despite all these limitations*, we do have interesting contributions and results for the NeurIPS community (and we sincerely thank Reviewers x6Jn and vTAU for voting toward acceptance). Namely:

**[Advances in meta-learning learning algorithms]**
Our meta-learned CL algorithm is successful at Split-MNIST unlike any hand-crafted algorithms and prior meta-learning methods for CL. This is a highly non-trivial achievement and a significant step in meta-learning research of learning algorithms. While Split-MNIST is indeed a toy task compared to other larger scale CL tasks (we also explicitly acknowledge this in the paper), it is not like the *standard* MNIST which is a *truly toy task* with many trivial solutions. We are not aware of any trivial solutions for Split-MNIST; it represents non-trivial CL challenges.

Regarding “practicality”, our main obstacle for showing results at scale is our compute resource limitation. While we do acknowledge in the paper certain technical challenges to be addressed in the future, much larger scale language modeling experiments are conducted by better-resourced teams.

**[Novel insights into in-context learning]**
Furthermore, the phenomena we exhibit and study, called “in-context catastrophic forgetting”, is a new perspective of in-context learning (ICL) that can be found nowhere in prior work. We largely dedicate space in our paper to discuss & analyze this in a comprehensible 2-task CL setting (Sec 4.1 and 4.2) using Mini-ImageNet and Omniglot. Please note that many recent foundational studies on ICL make use of “toy” tasks, such as regression to exhibit the core idea (see, e.g., [0]). We believe this is relevant to anybody in the NeurIPS community interested in ICL.

**[Novel results on sequence processing using fast weights and linear Transformer-family models]**
Finally, our work is also relevant to the readers interested in the general idea of sequence processing through “weight modifications” (also called fast weights). Such models are directly related to the linear-complexity variant of Transformers [1], and there is a very recent trend applying such models to language modeling (please see [2] and [3]). In particular, [2] shows that the model called DeltaNet [1] outperforms other popular linear-complexity models such as Mamba at scale. It should be noted that the direct followup of the DeltaNet [1, 2] is the SRWM architecture we use [4], which has been shown to be more expressive than the DeltaNet [4].  Therefore, going beyond the scope of continual and meta-learning, we believe our work also contributes to this emerging research on fast weight architectures for sequence processing [2, 3, 4] as another successful example thereof.

**For all these reasons**, we would really appreciate it a lot if the reviewers could take a look at our rebuttal response and reconsider the true contributions of this work through various angles/interests within the NeurIPS community.
We believe our responses provide convincing answers to all the main concerns raised by the reviewers (including a few crucial factual error corrections). We really believe our contributions outweigh the limitations, and we’ll be happy to provide further clarifications if necessary.

Once again, thank you very much for your valuable time reviewing our work.

References:

[0] von Oswald et al. ICML 2023. Transformers Learn In-Context by Gradient Descent. https://arxiv.org/abs/2212.07677

[1] Schlag et al. ICML 2021. Linear transformers are secretly fast weight programmers. https://arxiv.org/abs/2102.11174

[2] Yang et al. arXiv June 2024. Parallelizing Linear Transformers with the Delta Rule over Sequence Length.  https://arxiv.org/abs/2406.06484

[3] Sun et al. arXiv July 2024. Learning to (Learn at Test Time): RNNs with Expressive Hidden States. https://arxiv.org/abs/2407.04620

[4] Irie et al. EMNLP 2023. Practical Computational Power of Linear Transformers and Their Recurrent and Self-Referential Extensions. https://arxiv.org/abs/2310.16076

**== Correcting one factual error (Reviewers d5XY and HvJH) ==**

While we refer to our responses below for further individual clarifications, here we'd like to correct one factual error made by two reviewers.

> (Reviewer d5XY) Moreover, one limitation that the authors have not mentioned is that the meta-objective seems to require keeping in memory a number of copies of the model that is equal to the number of tasks.

> (Reviewer HvJH) In each step over a sequence of demonstrations, the method needs to compute and store a new weight matrix in order to perform back-propagation. It might require more memory during training and at inference.

These statements are not correct. No such storage/copies is required (there is a well known algorithm for linear Transformers to avoid this). Due to the rebuttal space limitation, we provide a detailed answer in our response to **Reviewer d5XY**.
We kindly ask **Reviewer HvJH** to refer to the corresponding text.

---

### Author Response · Authors · 2024-08-13
**Friendly request**

Dear Reviewers,

While we have been respecting the rule of not sending individual reminders to the reviewers ourselves, we would still like to share our honest feelings that we have found the author-discussion period very unfair so far.

Given the borderline ratings by three reviewers, we wrote a thorough rebuttal to resolve the concerns raised by all the reviewers.

In particular, we draw your attention to a few factual misunderstandings. For example,

Reviewer d5XY did not see the correct number of tasks used in the experiment ("showing an advantage of their approach in scenarios with up to 3 tasks"),

Reviewer HvJH overlooked comparisons to prior meta-learning methods,

and most critically, both Reviewers d5XY and HvJH raised concerns about a memory/storage problem of our training algorithms but any practitioner of linear Transformers knows that no such problem exists in practical implementations (if there were such an issue, it would have been impossible to run any experiments we ran): We find it unfair if we get rejected because of a problem that does not exist.

Influenced by these original reviews, Reviewer vTAU reduced the score from 6 to 4, which, in our view, lacks justifications as this decision ignores our rebuttal clarifications.

We completely understand that no single reviewer is expected to be fully familiar with all these technical aspects of this work, as we combine methods from various sub-fields of machine learning (continual learning, in-context/meta-learning, linear Transformers; and we believe this is the very strength of this work). However, we find that the current review/discussion status is unfair.

Regarding Reviewer x6Jn who seems to have thorough understandings of our work, we also addressed their remaining concern (regarding the impact of this work beyond our scope focused on SRWM) in our rebuttal.

We are respectfully waiting for justifications of their scores that take into account our rebuttal.

Thank you very much once again.

[We posted this message multiple times as OpenReview was experiencing some technical issues with notifications today. We deleted the old copies below.]

---

### Decision · Program_Chairs · 2024-09-25

**Decision:**

Reject

**Comment:**

This paper presents a meta continual learning method to deal with catastrophic forgetting. The method is based on ‘self-referential weight matrices’ (SRWM).
The key innovation is to define the loss function of SRWM training for optimizing both forward (improving performance of subsequent CL tasks) and backward (improving performance of previous CL tasks) transfer while achieving good performance on the current task.
Experiments demonstrate the effectiveness of the proposed method on common image classification meta-learning benchmarks such as Split-MNIST and Mini-ImageNet.

The paper received borderline reviews. While the reviewers acknowledged novel loss function, good number of experiments and good numeral results that demonstrated the proposed method outperforms the existing state of the art, there are shared concerns: the experimental setup is toy setting and is not scalable to a more practical one with a large number of tasks, the training process is expensive, and its performance is not as good as those based on pre-trained transformer models. In addition, it's unclear how to apply the proposed method to other tasks such as language modeling and multimodal tasks besides image classification ones. Overall, the reviewers were not convinced after the rebuttal.
 I think this paper is not ready for publication at the current stage.